# Optimal Design of Hydrometric Station Networks Based on Complex Network Analysis

Agarwal Ankit[1, 2, 3], Marwan Norbert[1], Maheswaran Rathinasamy[1], Ozturk Ugur [1, 2], Kurths Jürgen [1, 2,4], and Merz Bruno[2, 3]

[1]Research division Complexity Science, Potsdam Institute for Climate Impact Research, Member of the Leibniz Association, Telegrafenberg, Potsdam, Germany

[2]Institute for Environmental Sciences and Geography, University of Potsdam, Potsdam, Germany

[3]GFZ German Research Centre for Geosciences, Section 4.4: Hydrology, Telegrafenberg, Potsdam, Germany

[4]Institute of Physics, Humboldt Universität zu Berlin, Germany

[*]Corresponding author at: agarwal@pik-potsdam.de/ aagarwal@uni-potsdam.de

## Abstracts

Hydrometric networks play a vital role in providing information for decision-making in water resources management. They should be set up optimally to provide as much and as accurate information as possible, and at the same time, be cost-effective. Though, the design of hydrometric networks is a well-identified problem in hydrometeorology and has received considerable attention still it has scope for further advancement. In this study, we use complex network analysis, defined as collection of nodes interconnected by links, to propose a new measure that identifies critical nodes of station networks. The approach can support the design and redesign of hydrometric station networks. The science of complex networks is a relatively young field and has gained significant momentum in the last years in different areas such as brain networks, social networks, technological networks or climate networks. The identification of influential nodes in complex networks is an important field of research. In particular, we develop a new node ranking measure, the weighted degree-betweenness (WDB), to identify the stations providing the largest additional information to the network. The highest ranks of the WDB-ordered raingauges correspond to the most influential stations in the network. It is compared to previously proposed measures on synthetic sample networks and then applied to a real-world rain gauge network comprising 1229 stations across Germany to check its applicability in the optimal design of hydrometric networks. The proposed measure is evaluated using the decline rate of network efficiency and the kriging error. The results suggest that it effectively quantifies the importance of rain stations. The new measure is very useful in identifying influential stations which need high attention and expendable stations which can be removed without much loss of information provided by the station network.

Keywords: Rainfall network, complex networks, event synchronization, kriging error.

## 1 Introduction

Hydrometric networks monitor a wide range of water quantity and water quality parameters such as precipitation, streamflow, groundwater, or surface water temperature (Keum et al., 2017). Adequate hydrometric monitoring is one of the

first and primary tasks towards efficient water resources management. Information from hydrometric stations plays a crucial role in, among other things, flood estimation, water budget analysis, hydraulic design and assessing climate change. Even after the advent of remote sensing based information, such as precipitation products, in-situ observations are considered as an essential source of information in hydrometeorology.

The basic characteristics of hydrometric networks comprise the number of stations, their locations, observation periods and sampling frequency (Keum et al., 2017). The general understanding is that the higher the number of monitoring stations, the more reliable the quantification of areal average estimates and point estimates at any ungauged location. However, a higher station number increases the cost of installation, operation, and maintenance, but may provide redundant information and, therefore, not increase the information content obtained from the network. Globally, there is a decreasing trend in the
number of hydrometric stations in the last decades (Mishra and Coulibaly, 2009). Against the background of shrinking monetary support for hydrometric networks, their optimal design is gaining importance.

The design of hydrometric networks is a well-identified problem in hydrometeorology and has received considerable attention (Mishra and Coulibaly, 2009). For example, Putthividhya and Tanaka (2012) made an effort to design an optimal rain gauge network based on the station redundancy and the homogeneity of the rainfall distribution. Adhikary et al. (2015)
proposed a kriging based geostatistical approach for optimizing rainfall networks, and Chacon-Hurtado et al. (2017) provided a generalized procedure for optimal rainfall and streamflow monitoring in the context of rainfall-runoff modelling. Yeh et al. (2017) optimized a rain gauge network applying the entropy method on radar datasets. Several approaches have been developed for optimal network design, such as statistical analysis which include variance and dimension reduction methods (Wadoux et al., 2017), spatial interpolation which includes kriging methods (Adhikary et al., 2015) and various
interpolation techniques (Kassim and Kottegoda, 1991), information theory-based methods (Stosic et al., 2017), optimization techniques such as simulated annealing (Mishra and Coulibaly, 2009), physiographic analysis (Laize, 2004), multivariate factor analysis (Hargrove and Hoffman, 2004), sampling strategies (Tsintikidis et al., 2002), and user surveys or expert recommendations (Rani and Moreira, 2010). Combinations of methods have also been introduced in the last decade (Chacon-Hurtado et al., 2017; Keum et al., 2017; Mishra and Coulibaly, 2009).

Most of these studies inherently assume that a more optimal network is achieved through expanding the network with supplementary stations. However, increasing the number of stations does not necessarily decrease the uncertainty (Stosic et al., 2017). Mishra and Coulibaly (2009) argued that the expendable stations in a network that contribute little or even nothing should be identified and removed, and at the same time, the most valuable or influential stations should be maintained and protected. Hence, a network can also be optimized by eliminating expendable stations from the network.

Against this background, this study aims to identify influential and expendable stations based on their relative information content by developing a new node ranking measure for hydrometric station networks. We use complex network which is defined as a collection of nodes interconnected with links in a non-trivial manner. The application of complex networks in hydrology is still in its infancy stage, however it has attracted many researchers from different disciplines and application

fields, e.g., transportation networks (Bell and Lida, 1997), power grid analysis (Schultz et al., 2014), streamflow networks (Halverson and Fleming, 2015) and climate networks (Agarwal et al., 2018b).

We use complex networks since it is a powerful approach in extracting information from large high-dimensional hydrological datasets (Donges et al., 2009a; Cohen and Havlin 2010). This non-parametric method allows investigating the topology of local and non-local statistical interrelationships. An example for non-local connections in a climate network are global influence of El Nino Southern Oscillation (ENSO) on rainfall (Agarwal, 2019; Ferster et al., 2018) and Atlantic Meridional Overturning Circulation (AMOC) on air surface temperature (Caesar et al., 2018) via teleconnections and ocean circulation respectively. The method allows to represent the dataset in form of spatially embedded network and visualize the connections. Once the spatial network of stations is set up, one might use network measures (e.g. degree, betweenness centrality) to analyse a range of aspects, such as community structure unravelling dominant climate modes (Agarwal et al., 2018a; Fang et al., 2017; Halverson and Fleming, 2015; Tsonis et al., 2011), catchment classification indicating hydrologic similarity (Fang et al., 2017), short and long-range spatial connections in rainfall (Agarwal et al., 2018a; Boers et al., 2014b; Jha et al., 2015; Stolbova et al., 2014) and spatio-temporal hydrologic patterns (Halverson and Fleming, 2015; Konapala and Mishra, 2017). Further, a recent study by Donges et al., (2015) pinpoints that complex network analysis can complement classical Eigen techniques, such as empirical orthogonal functions (EOFs) or coupled patterns (CP) maximum covariance analysis. They showed that EOFs, CPs and related methods rely on dimensionality reduction, whereas network techniques allow studying the full complexity of the statistical interdependences structure and are not limited to linear and spatial-proximity connections. Also, it has been shown that higher-order complex network measures (betweenness centrality, closeness centrality, participation coefficient) provide additional information on the higher-order structure of statistical interrelationships in climatological data (Donges et al., 2015). For example, the network degree represents similar information as the first eigenmode of an EOF analysis (Donges et al, 2015), in-out degree (incoming links and outgoing links in directed network) represents spatial-temporal propagation (Boers et al., 2014a), and betweenness centrality represents a spatial separation between flows or a handful of stations which are positioned in-between the large communities. These stations belong to large communities, but unlike most stations they tend to possess intercommunity connections ((Halverson and Fleming, 2015; Molkenthin et al., 2015; Tupikina et al., 2016).

In this study, we propose a complex network-based method to identify the influential and expendable stations in a rainfall network. The novelty of this study is twofold: 1) We propose a new measure for identifying the most influential nodes in a network, and 2) we use event synchronization as a similarity measure. Several methods in the field of complex networks have been proposed to evaluate the importance of nodes (Chen et al., 2012; Hou et al., 2012; Jensen et al., 2016; Kitsak et al., 2010; Zhang et al., 2013 and Hu et al., 2013). Degree (k), betweenness centrality (B) and closeness centrality (CC) are the methods commonly used in complex networks (Gao et al., 2013). Studies in different disciplines have shown that degree and betweenness centrality often outperform other node-ranking measures (Gao et al., 2013; Liu et al., 2016). We propose a novel measure, weighted degree-betweenness (WDB), which combines degree (k) and betweenness centrality (B), to identify

the stations providing the largest information to the network. We show that the proposed measure WDB has a higher discrimination power compared to existing methods and that it effectively ranks the nodes in the network. Additionally, WDB is more sensitive to the different roles of nodes, such as global connecting nodes, hybrid nodes, and local centres, and provides a more informative ranking than the existing node ranking measures.

Further, we use event synchronization as a similarity measure. In a complex network, links are set up between each pair of nodes based on how the nodes interact with each other. This interaction is measured through statistical measures, such as zero-lag correlation or time-delayed correlation (Agarwal et al., 2018b). However, these measures are limited by the underlying assumptions, e.g. measuring linear relations. They give equal weight to high and low rainfall values, whereas the main information content in a rainfall time series is embedded in the larger values. In contrast, event synchronization (ES) is

a suitable measure for event-like, non-Gaussian data such as precipitation (Malik et al., 2012; Tass et al., 1998). It has advantages over other time-delayed correlation techniques (e.g., Pearson lag correlation), as it allows us to define the event time series by determining the threshold, and as it uses a dynamic time delay (not fixed). The latter refers to a time delay that is adjusted according to the two time series being compared, which allows for better adaptability to the variable and region of interest.

The main objective of the study is to develop a node ranking measure, based on complex network analysis that can be used to identify influential and expendable stations in large hydrometric station networks. Our aim is not to question the credibility of operating stations, but to propose an alternative evaluation procedure for the optimal design and redesign of observational networks.

In section 2, we introduce the basic concepts of complex networks. The proposed node ranking measure is presented and

compared with existing measures in section 3 using synthetic networks. In section 4, the new measure is applied to a rain gauge network consisting of 1229 stations across Germany and compared with state-of-the-art methods.

## 2 Basics of Complex Networks

### 2.1 Network Construction

A network or a graph is a collection of entities (nodes, vertices) interconnected with lines (links, edges) as shown in Fig. 1.

These entities could be anything, such as humans defining a social network (Arenas et al., 2008), computers constructing a web network (Zlatić et al., 2006), neurons forming brain networks (Bullmore and Sporns, 2012), streamflow stations creating a hydrological network (Halverson and Fleming, 2015) or climate stations describing a climate network (Agarwal et al., 2018a).

Formally, a network or graph is defined as an ordered pair $Z = \{N, E\}$; containing a set $N = \{N_1, N_2, .... N_N\}$, of vertices

together with a set $E$ of edges, $\{i, j\}$ which are 2-element subsets of N. In this work we consider undirected and unweighted

simple graphs, where only one edge can exist between a pair of vertices and self-loops of the type $\{i,i\}$ are not allowed. This type of graph can be represented by the symmetric adjacency matrix (Fig.1)

$$A_{i,j} = \begin{cases} 0 & \{i,j\} \notin E \\ 1 & \{i,j\} \in E \end{cases} \tag{1}$$

Figure 1 is a simple representation of such a network, i.e., one with a set of identical nodes connected by identical links. In general, (large) graphs of real-world entities with irregular topology are called complex networks. The links represent similar evolution or variability at different nodes and can be identified from data using a similarity measure such as Pearson correlation (Donges et al., 2009a), synchronization (Agarwal, 2019; Boers et al., 2019; Conticello et al., 2018) or mutual information (Paluš, 2018).

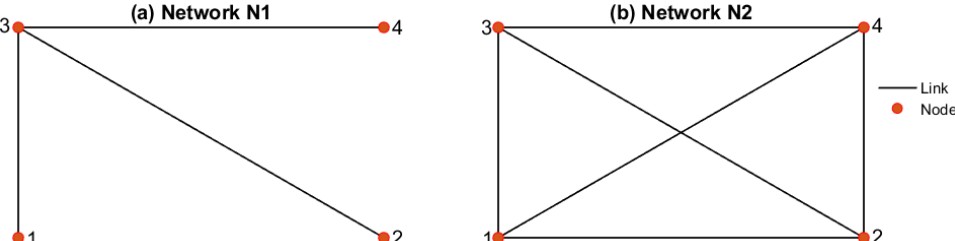

**Figure 1: Topology of two sample networks to explain network structures and measures. (a) Network N1 with four nodes and three links; (b) network N2 with four nodes and six links.**

## 2.2 Event synchronization

Event synchronization (ES) has been specifically designed to calculate nonlinear correlations among bivariate time series with events defined on them (Quiroga et al., 2002). This method has advantages over other time-delayed correlation techniques (e.g., Pearson lag correlation), as it allows us to define extreme event series of the signal, depending on the kind of extreme, and as it uses a dynamic time delay. The latter refers to a time delay that is adjusted according to the two time series being compared, which allows for better adaptability to the variable and region of interest. Another advantage of this method is that it can also be applied to a non-Gaussian and event-like data sets (Boers et al., 2014b, 2015; Malik et al., 2012).

In the last decade, various modifications have been proposed, related to, for instance, boundary effects (Rheinwalt et al., 2016) and bias toward the number of events which can be explained as, let us say an event above threshold $\alpha$ percentile occurs in the signal $x(t)$ and $y(t)$ (Fig. 2, step 1) at time $t_l^x$ and $t_m^y$ where $l = 1,2,3,4 \dots S_x$, $m = 1,2,3,4 \dots \dots S_y$ and within a time lag $\pm \tau_{lm}^{xy}$ which is defined as following (Fig. 2, step 2)

$$\tau_{lm}^{xy} = min\{t_{l+1}^x - t_l^x, t_l^x - t_{l-1}^x, t_{m+1}^y - t_m^y, t_m^y - t_{m-1}^y \}/2 \tag{2}$$

where $S_x$ and $S_y$ is the total number of such events (greater then threshold $\alpha$) that occurred in the signal $x(t)$ and $y(t)$, respectively. The above definition of the time lag helps to separate independent events which in turn allows to take into

account the fact that different processes may be responsible for the generation of events. We need to count the number of times an event occurs in the signal $x(t)$ after it appears in the signal $y(t)$, and vice versa, and this is achieved by defining quantities $C(x|y)$ and $C(y|x)$ where

$$C(x|y) = \sum_{l=1}^{S_x} \sum_{m=1}^{S_y} J_{xy} \tag{3}$$

and

$$J_{xy} = \begin{cases} 1 & if \quad 0 < t_l^x - t_m^y < \tau_{lm}^{xy} \\ \frac{1}{2} & if \quad t_l^x = t_m^y \\ 0 & else, \end{cases} \tag{4}$$

This definition of $J_{xy}$ prevents counting a synchronized event twice. When two synchronized events match exactly ($t_l^x = t_m^y$), we use a factor $1/2$ since they double count in $C(x|y)$ and $C(y|x)$. Similarly, we can define $C(y|x)$ and from these quantities we obtain

$$Q_{xy} = \frac{C(x|y) + C(y|x)}{\sqrt{(S_x - 2)(S_y - 2)}} \tag{5}$$

$Q_{xy}$ is a normalized measure of the strength of event synchronization between signal $x(t)$ and $y(t)$. This implies $Q_{xy} = 1$ for perfect synchronization and zero if no events are synchronized.

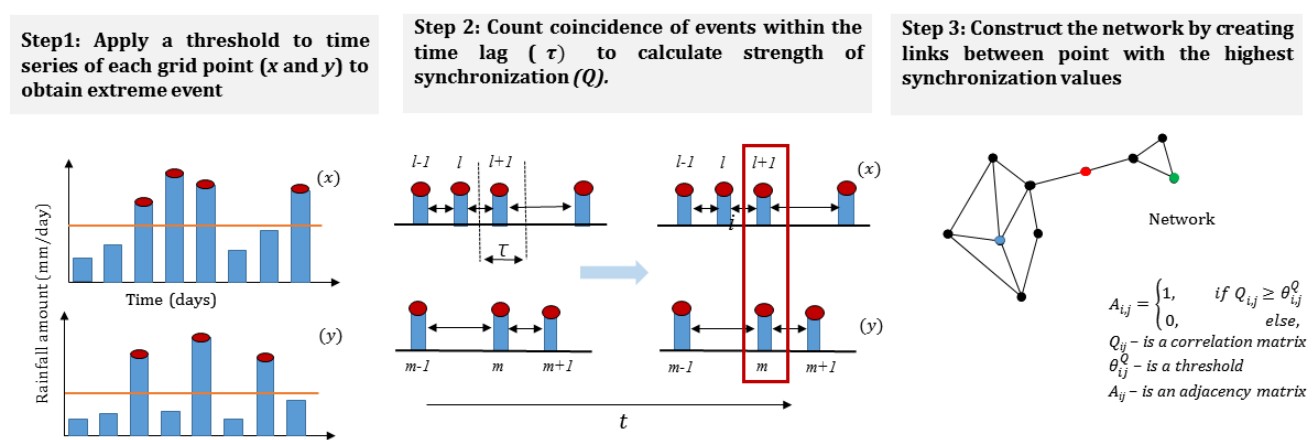

Figure 2: Schematic of network construction using event synchronization (ES). Equations and symbols are given in the main text.

## 2.3 Node Ranking Measures

A large number of measures have been defined to characterize the behaviour of complex networks. We focus here on those traditional and contemporary network measures which have been proposed to quantify the importance of nodes in a network:

degree *k,* betweenness centrality *B* (Stolbova et al., 2014), bridgeness *Bri* (Jensen et al., 2016) and degree and influence of line *DIL* (Liu et al., 2016).

### *Traditional network measures*

The degree *k* of a node in a network counts the number of connections linked to the node directly. The degree of any *i* node is calculated as

$$k_i = \frac{\sum_{j=1}^{N} A_{i,j}}{N-1} \tag{6}$$

Where *N* is the total number of nodes in a network. For example, the degree of nodes 1, 2 and 4 in network N1 (Fig. 1a) is 1 and for node 3 is 3. In the network N2 (Fig. 1b), all nodes have degree 3. The degree can explain the importance of nodes to some extent, but nodes that own the same degree may not play the same role in a network. For instance, a bridging node connecting two important nodes might be very relevant though its degree could be much lower than the value of less important nodes.

The betweenness centrality *B* is a measure of control that a particular node exerts over the interaction between the remaining nodes. In simple words, *B* describes the ability of nodes to control the information flow in networks. To calculate betweenness centrality, we consider every pair of nodes and count how many times a third node can interrupt the shortest paths between the selected node pair. Mathematically, betweenness centrality *B* of any *i* node is

$$B_i = \sum_{i \neq j \neq v \in \{V\}}^{N} \frac{\sigma_i(j,k)}{\sigma(j,k)} \tag{7}$$

where $\sigma(j,k)$ represents the number of links along the shortest path between node *j* and *k*; while $\sigma_i(j,k)$ is the number of links of the shortest path running through node *i*. In network N1, *B* of node 3 is 3, i.e., node 3 can disturb three pairs 1-2, 1-4, 2-4, and for other nodes $B = 0$. In the network N2, all nodes have $B = 0$ because no node can interrupt the information flow. So node 3 is a critical node in the network N1 but not in the network N2.

### *Contemporary network measures*

Jensen et al. (2016) developed the Bridgeness measure *Bri* to distinguish local centres, i.e. nodes that are central to a part of the network, from hybrid nodes, i.e. nodes that connect different parts of a network (Fig. 3). *Bri* is a decomposition of betweenness centrality *B* into a local and a global contribution. Therefore, the *Bri* value of a node *i* is always smaller or equal to the corresponding *B* value and they only differ by the local contribution of the first neighbours. To calculate *Bri* we consider the shortest path between nodes outside the neighbourhood of node $i, N_G(i)$. Mathematically, it is represented as

$$Bri_i = \sum_{j \notin N_G(i) \lor k \notin N_G(i)}^{N} \frac{\sigma_i(j,k)}{\sigma(j,k)} \tag{8}$$

The neighbourhood of node $i$ ($N_G(i)$) consists of all direct neighbours of node $i$. For example, in the networks N1 and N2, all nodes have $B = 0$, hence $Bri = 0$, except node 3 in the network N1 for which all the nodes are in direct neighbourhood. Hence, it also has $Bri = 0$.

The degree and influence of line ($DIL$), introduced by Liu et al. (2016), considers the node degree $k$ and importance of line $I$ to rank the nodes in a network:

$$DIL_i = k_i + \sum_{j=N_G(i))} I_{e_{ij}} \cdot \frac{k_i - 1}{k_i + k_j - 2} \tag{9}$$

where the line between node $i$ and j is $e_{ij}$ and its importance is defined as $I_{e_{ij}} = \frac{U}{\lambda}$ where $U = (k_i - p - 1).(k_j - p - 1)$ reflects the connectivity ability of a line (link), $p$ is the number of triangles having one edge $e_{ij}$ and $\lambda = \frac{p}{2} + 1$ is defined as an alternative index of line $e_{ij}$. $N_G(i))$ is the set of neighbours of node $i$ (for detailed explanation refer Liu et al., 2016). The equation for $DIL$ suggests that all the nodes having $k_i = 1$ will have $DIL_i = 1$, since the second term of the equation will be zero. Hence, in the network N1 all nodes, except node 3, have $DIL = 1$. Node 3 has $DIL = 3$ equal to its degree, since the second term is zero (all the connected nodes 1, 2 and 4 have $k_j = 1$, hence $I_{e_{ij}} = 0$).

## 3 Methodology

We propose a new node ranking measure that we call weighted degree-betweenness (WDB). We further compare the efficacy of the proposed measure with the existing traditional and contemporary node ranking methods using two synthetic networks.

### 3.1 Weighted Degree-Betweenness

WDB is a combination of the network measures degree and betweenness centrality. We define WDB of a particular node $i$ as the sum of the betweenness centrality of node $i$ and all directly connected nodes $j, j = 1,2,3 ....n$ in proportion to their contribution to node $i$. Mathematically, the WDB of a node $i$ is given by

$$WDB_i = B_i + I_i \tag{10}$$

*where $B_i$* is the betweenness centrality of node $i$, and $I_i$ stands for the influence or contribution of the directly connected node j, $j = 1,2,3 ....n$ to node $i$. It is defined for node $i$ as

$$I_i = \sum_{j=1}^{n} \frac{B_j * (k_j - 1)}{(k_i + k_j - 2)} \tag{11}$$

*where $k_i$* is the degree of node i, $k_j$ is the degree of the nodes j which are directly connected to node $i_i$, and $n$ is the total number of directly connected nodes to node $i$.

**3.2 Comparison with Existing Node Ranking Measures Using Synthetic Networks**

In this section, we motivate the development of the new node ranking measure WDB by comparing it to existing network measures. Identifying nodes that occupy interesting positions in a real-world network using node ranking helps to extract meaningful information from large datasets with little cost.

Usually, the measures degree or betweenness centrality are used for node ranking (Gao et al., 2013; Okamoto et al., 2008; Saxena et al., 2016). However, these measures have certain limitations which are explained using a simple network, the undirected and unweighted network $Z = (N, E)$ with 8 nodes and 11 edges shown in Figure 3. The network measures $k_i$, $B_i$ and $WDB_i$ of each node are given in Figure 3 along with the node number.

Degree is limited as node ranking measure since it cannot distinguish between different roles in the network. For example, nodes 5, 7, and 8 have the same degree ($k_i$=2), but node 5 serves as a bridge node linking the two parts of the network. Information between several nodes in this network can flow through this node only. In a large complex network, such nodes have strategic relevance as most of the information can be accessed quickly just by capturing those nodes. For example, in a social network, the spreading of a disease could be slowed down or hindered by identifying these nodes. In climate networks, an early warning signal could be generated by capturing the flow of information (Donges et al., 2009a, 2009b).

Betweenness centrality has a higher power in discriminating different roles. For example, nodes 4 and 5 have the highest betweenness centrality $B = 24$ followed by node 6. Their importance for the information flow in the network is obvious, as such high $B$ nodes can be used to control the flow of information in any network. Information flow, or in our context, transferability of precipitation measurements across locations, would be restricted in their absence. However, betweenness $B$ gives equal scores to local centers (nodes 4, 6), i.e., nodes of high degree central to a single region, and to global bridges (node 5), which connect different regions. This distinction is important because the roles of these nodes are different. For example, in climate networks, local centers correspond to nodes which are important for local climate phenomena, while bridges correspond to nodes which connect different climatic subsystems, such as Indian monsoon and El Niño/Southern Oscillation, leading to teleconnections (Paluš, 2018). Bridge nodes spread a process to the entire region globally whereas the effect of a local center is confined to a region (Lawyer, 2015).

In climate networks, information and/or mass can be transported between nodes. In temperature-based climate networks, it is the energy that is transported, and with this, some kind of information about the atmospheric state in a region (Hlinka et al., 2017). For rainfall networks, the links may reflect the major propagation pathways of moisture (Boers et al., 2013, 2014b; Malik et al., 2012). For extreme precipitation, it is even more specific and reflects certain weather conditions, e.g. a specific weather pattern in central Europe as shown by Rheinwalt et al., (2016). Ozturk et al., (2018) proposed a complex network-based approach to estimate the movement of extreme rainfall over Japan during typhoons. They iteratively estimated likely tracks of extreme precipitation for each cell of a large grid, many of which present redundant information, and hence the computation is time consuming. We suggest that by applying their method only on global bridges and local centers, we can deduce the likely tracks of extreme rainfall efficiently.

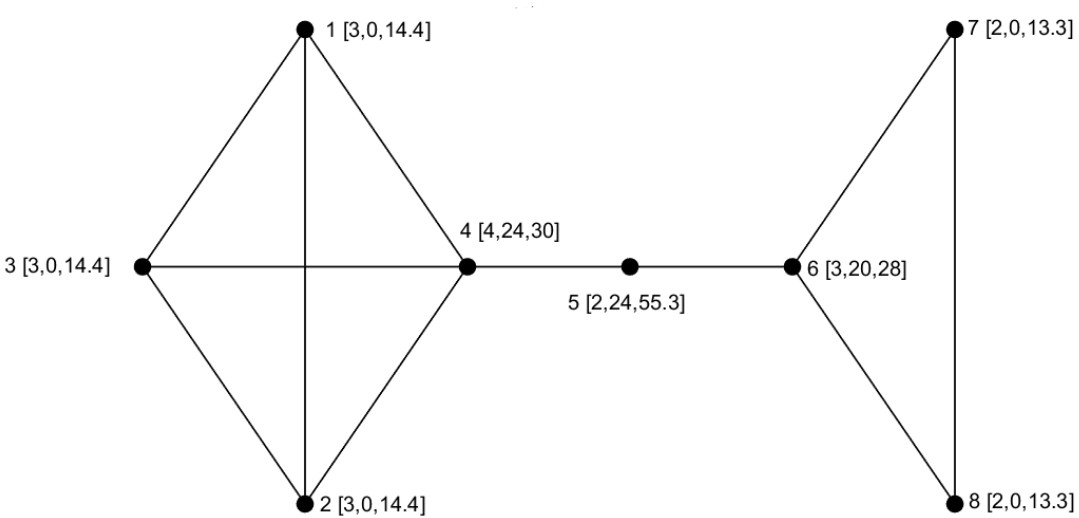

**Figure 3: Synthetic network to explain the degree *(k)*, betweenness (*B*) and weighted degree-betweenness (*WDB*) measures, with node number (1 to 8) followed by the degree, betweenness value and *WDB* value in brackets. Degree and betweenness are limited in distinguishing the role of different nodes in the network and centers from bridges, respectively.**

5    The proposed measure *WDB* has an even higher discrimination power compared to betweenness centrality and effectively ranks the nodes in the network. Node 5 has the highest *WDB* score and is ranked as the most influential node. This reflects its role as global bridge node, as losing node 5 would disconnect the two parts of the network. *WDB* is also able to distinguish between the nodes 1, 2, 3 (*WDB* = 14.4) and the nodes 7, 8 (*WDB* = 13.3), which is important in case we need to sequentially rank nodes.

10  To further evaluate the proposed measure, we compare *WDB* with other network measures recently published, namely the bridgeness developed by Jensen et al. (2016) and degree and influence of line DIL by Liu et al. (2016). For this comparison, we use the same synthetic network as Jensen et al. (2016) shown in Fig. 4. The corresponding network measure values are also given in Fig. 4.

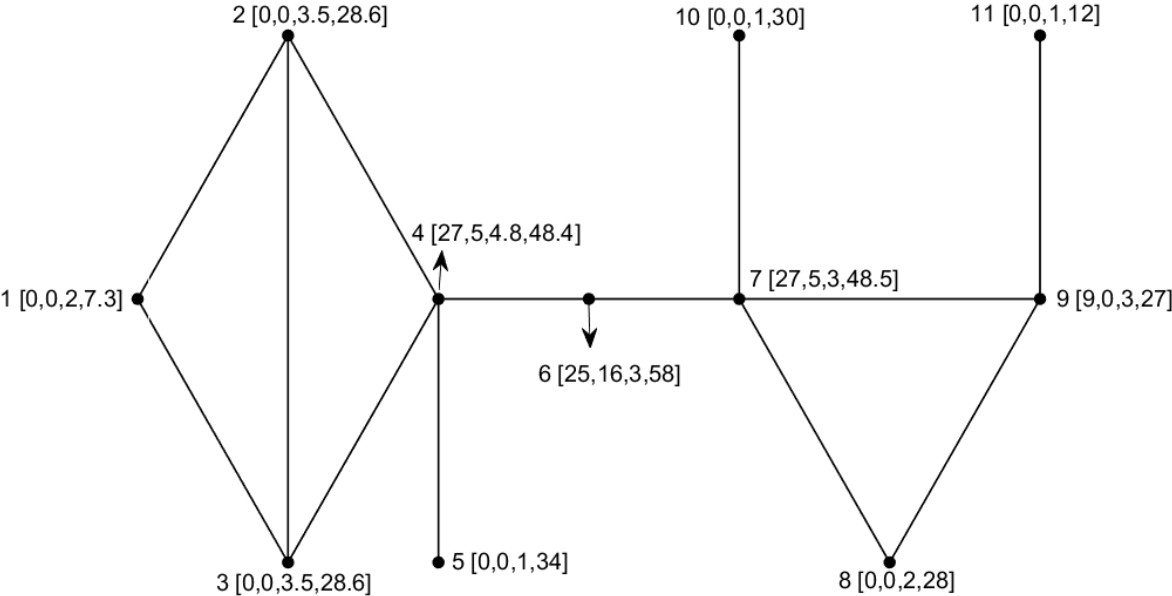

**Figure 4: Synthetic network used to compare the network measures betweenness, bridgeness, and DIL with the proposed measure WDB. Numbers 1 to 11 are node counts, and values in brackets represent the network measure values in order of $[B, Bri, DIL, and\ WDB]$. Node 6 is a global bridge node which connects two sub-networks. Node 4 and 7 are hubs which are**
**connected to most of the nodes in the sub-networks. Node 5, 10 and 11 are the dead-end nodes.**

Fig. 4 illustrates that betweenness does not distinguish between the local centers (nodes 4, 7) and the global bridge node (node 6). It even assigns a smaller value to the global bridge node. Bridgeness expresses the higher importance of the global bridge node compared to local centers, however, it does not distinguish between all other nodes. Although *DIL* assigns different values to almost very node, these numbers do not represent the different roles of the nodes and are therefore hardly

suitable as node ranking measure. *WDB* outperforms the existing measures in effectively ranking nodes in the network, such as the global bridge nodes, local centers and dead-end nodes. For example, WDB differentiates between nodes 4 and 7 for which the bridgeness measure provides equal scores.

This comparison of the proposed measure *WDB* with other measures that have been developed to express the importance of nodes within a network shows that WDB is able to provide a nuanced picture. The resulting node ranking reflects the

different roles, such as global bridge, local center, dead-end node, hub (high degree), or non-hub (low degree), of the individual nodes.

There is one situation where our method would require additional care: Let us imagine a node that is unrelated to other nodes (no links). One might imagine this scenario in a meteorological sub-region characterized by fine-scale convective thunderstorms with sparse rain gauge coverage. Hence, precipitation event synchronization in that sub-region would be

poor. This station would not be the part of the constructed network and would not be ranked. However, this station should be treated carefully as it provides unique information.

## 3.3 Evaluation of the Proposed Measure for a Rain Gauge Network

In the context of hydrometric station networks, we hypothesise that higher ranking stations are more influential nodes in the network. Loosing such stations would more strongly reduce the network efficiency, i.e., the flow of information within the network, compared to lower ranking stations. Stations with the lowest ranks in the network are the least influential and are seen as expendable stations. To test this hypothesis, we apply the proposed node ranking measure to a rain gauge network consisting of more than 1000 stations in Germany. The information loss caused by removing stations is quantified via two measures: (a) decline rate of network efficiency, and (b) relative kriging error.

### 3.3.1 Decline Rate of Network Efficiency

The decline rate of network efficiency, as proposed by Liu et al. (2016), quantifies the loss in efficiency with which information flows within a network when nodes are removed from the network. Network efficiency is defined as

$$\eta = \frac{1}{N(N-1)} \sum_{n_i \neq n_j} \eta_{ij} \tag{12}$$

Where $N$ is the total number of nodes in a network. $\eta_{ij}$ is the efficiency between nodes $n_i$ and $n_j$. $\eta_{ij}$ is inversely related to the shortest path length: $\eta_{ij} = 1/d_{ij}$, where $d_{ij}$ is the shortest path between nodes $n_i$ and $n_j$. The average path length $L$ measures the average number of links along the shortest paths between all possible pairs of network nodes. It is a measure of the efficiency of information or mass transport in a network. A network with small $L$ is highly efficient, because two nodes are likely to be separated by a few links only. The decline rate of network efficiency $\mu$ is defined as

$$\mu = 1 - \frac{\eta_{new}}{\eta_{old}} \tag{13}$$

where $\eta_{new}$ is the efficiency of the network after removing nodes, and $\eta_{old}$ is the efficiency of the complete network.

We hypothesise that the network efficiency reduces more strongly if higher ranking stations are removed. This implies higher decline rates of efficiency when removing higher ranking stations from the network.

### 3.3.2 Relative Kriging Error

As second measure to evaluate the information loss when stations are removed from the network, we use a kriging based geostatistical approach (Adhikary et al., 2015; Keum et al., 2017). Kriging is an optimal surface interpolation technique assuming that the variance in a sample of observations depends on their distance (Adhikary et al., 2015). It is the best linear unbiased estimator of unknown variable values at unsampled locations in space where no measurements are available, based on the known sampling values from the surrounding areas (Hohn, 1991; Webster and Oliver, 2007). Ordinary Kriging is used in this study for interpolating rainfall data and estimating the kriging error. The kriging estimator is expressed as

$$Z^*(x_o) = \sum_{i=1}^{n} w_i Z(x_i) \tag{14}$$

where $Z^*(x_0)$ refers to the estimated value of Z at the desired location $x_0$; $w_i$ represents weights associated with the observation at the location $x_i$ with respect to $x_0$; and n indicates the number of observations within the domain of the search neighborhood of $x_0$ for performing the estimation of $Z^*(x_0)$. Ordinary Kriging is implemented through ArcGISv10.4.1 (Redlands, CA, USA) (ESRI, 2009) and its geostatistical analyst extension (Johnston et al., 2001).

The kriging variance $\sigma_z^2(x_o)$ in the Ordinary Kriging can be computed as (Adhikary et al., 2015; Xu et al., 2018)

$$\sigma_z^2 = \mu_z + \sum_{i=1}^{n} w_i \gamma(h_{oi}) \quad \text{for} \sum_{i=1}^{n} w_i = 1$$

where $\gamma(h)$ is the variogram value for the distance h; $h_{0i}$ is the distance between observed data points $x_i$ and $x_j$; $\mu_z$ is the Lagrangian multiplier in the Z scale; $h_{0j}$ is the distance between the unsampled location $x_0$ (where the estimation is desired) and sample locations $x_i$; and n is the number of sample locations.

The square root of the kriging variance, also named as kriging standard error (KSE), is used as a gauge network evaluation factor. We estimate the increase in the kriging standard error across the study area when stations are removed to evaluate the performance of the WDB measure in identifying influential and expendable stations in a large network. Goovaerts (1997, p. 179) states that the theoretical kriging error is dependent on the variogram model and data configuration whereas it is independent of data values (Appendix B). In a given scenario of constant variogram during network modifications

theoretical kriging error only depends on data configuration (density and distribution of inter-distances between stations).

The relative kriging error before and after removing the stations is denoted as

$$\Re(\%) = \frac{KSE_{new} - KSE_{old}}{KSE_{old}} \times 100 \tag{15}$$

where $KSE_{new}$ denotes the standard kriging error after removing stations, and $KSE_{old}$ is the error for the original network. We hypothesise that the increase in the relative kriging error is higher when removing high ranking stations. To cover a broad range of rainfall characteristics, the error is calculated for different statistics, i.e. the mean, 90[th], 95[th] and 99[th] percentile

rainfall and the number of wet days (precipitation > 2.5mm).

**4 Application to an Extensive Rain Gauge Network**

**4.1 Rainfall Data**

To evaluate the proposed measure in the context of the optimal design of hydrometric networks, we apply it to an extensive network of rain stations in Germany and adjacent areas (Figure 5). The data covers 110 years at daily resolution (1 January

1901 to 31 December 2010). The 1229 rain stations inside Germany (blue dots in Fig. 5) are operated by the German Weather Service. Data processing and quality control were performed according to Österle et al. (2006), and in this study we

assume that data is free from measurement errors. 211 stations from different sources outside Germany (red dots in Fig. 5) were included in the analysis to minimize spatial boundary effects in the network construction, however, these stations were excluded from the node ranking analysis.

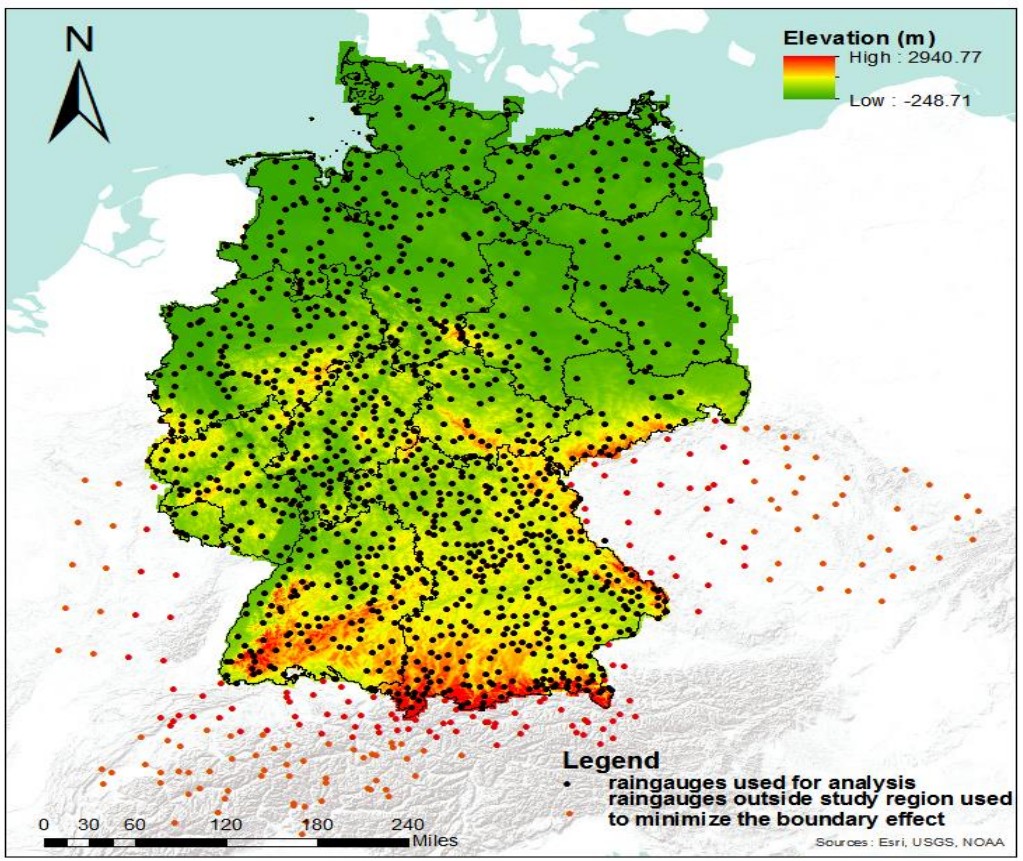

**Figure 5: Location of rain stations in Germany and adjacent areas. Blue dots indicate stations lying inside Germany that are used in the analysis. Red dots indicate stations outside of Germany that are used for network construction only to minimize the boundary effect.**

### 4.2 Network Construction

We begin the network construction by extracting event time series from the 1440 daily rainfall time series. The event series represent heavy rainfall events, i.e., precipitation exceeding the 95th percentile at that station (Rheinwalt et al., 2016). The 95th percentile is a good compromise between having a sufficient number of rainfall events at each location and a rather high threshold to study heavy precipitation. All rainfall event series are compared with each other using event synchronization (Fig. 2) which is the base for various network measures used to rank the nodes in the network. Hence, event synchronization is not used to derive the station ranking. This results in the similarity matrix Q, whereas the entry at index pair ($i,j$) defines

synchronization in the occurrence of heavy rainfall events at station $i$ and station $j$. Applying a certain threshold to the $Q$ matrix (see Appendix A) yields an adjacency matrix (Fig. 2)

$$A_{i,j} = \begin{cases} 1, & if\ Q_{i,j} \geq \theta_{i,j}^Q \\ 0, & else, \end{cases} \tag{16}$$

Two criteria have been proposed to generate an adjacency matrix from a similarity matrix, such as fixed amount of link density (Agarwal et al., 2018a; Donges et al., 2009a; Stolbova et al., 2014) or global fixed thresholds (Jha et al., 2015; Sivakumar and Woldemeskel, 2014). However, both criteria are subjective and may lead to the presence of weak and non-significant links in the adjacency matrix or network. These non-significant links might obscure the topology of strong and significant connections, Hence, stringent threshold criteria are needed, such as multiple testing (Agarwal, 2019; Boers et al., 2019). Alternatively, networks should be characterized across a broad range of thresholds. Furthermore, all self-connections or negative connections (anti-correlation), if any, should be removed (Rubinov and Sporns, 2010).

To minimize these threshold effects, we choose the threshold $\theta_{i,j}^Q$ objectively by considering all links in the network that are significant. A link is significant (i.e. two stations are significantly synchronized) if the synchronization value exceeds the 95[th] percentile of the synchronization obtained by two synthetic variables that have the same number of events positioned randomly in the time series. We calculate synchronization for 100 pairs of random time series from which we derive the 95[th] percentile of synchronization. Using a 5% significance level, we assume that synchronization cannot be explained by chance, if the ES value between two stations is larger than the 95[th] percentile of the test distribution. Here, we select 5% significance level since it is the well accepted criteria for the network construction.

$A_{i,j} = 1$ denotes a link between the $i^{th}$ and $j^{th}$ station and 0 denotes otherwise. The adjacency matrix represents the connections in the rainfall network. Although the constructed network is based on all 1440 stations (to minimize the boundary effect), the subsequent topological analysis is performed only for the 1229 stations lying inside Germany.

## 4.3 Decline Rate of Network Efficiency

In this section, we evaluate the ranking of stations derived from the proposed WDB measure using the decline rate of network efficiency. The rain gauges are ranked in decreasing order according to their WDB values. Highly ranked rain gauges are interpreted as the most influential stations, and low ranked as expendable stations.

Firstly, we analyze the decline rate of network efficiency $\mu$ when one station is removed from the network. In each trial, we remove only one station (starting with the highest rank). After $n=1229$ (number of nodes) trials, we investigate the relationship between $\mu$ and the node ranking measured by WDB. We expect an inverse relationship between $\mu$ and WDB: the higher the node ranking, the more important is that node, leading to a higher loss in network efficiency. Fig. 6 confirms this behavior. $\mu$ is high for high-ranking stations and decays with node ranking. Interestingly, $\mu < 0$ for very low ranking stations, i.e. the network efficiency increases when single, low ranking stations are removed. This is explained by the decrease of the redundancy in the network when such stations are removed.

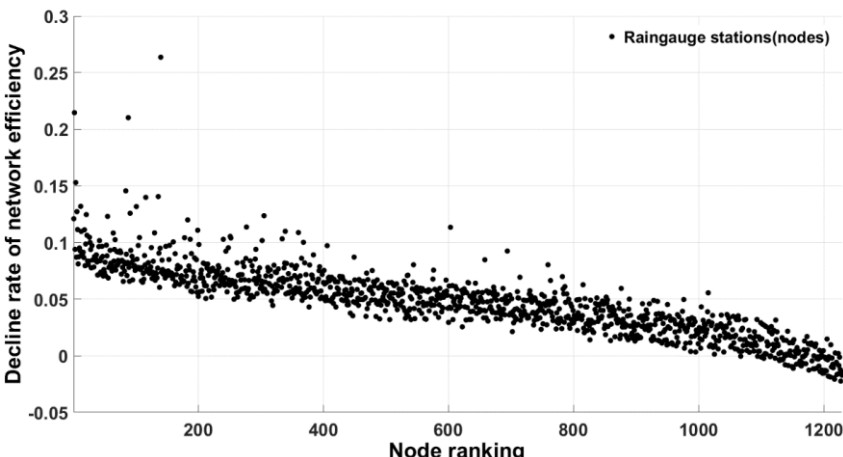

**Figure 6: Decline rate of network efficiency corresponding to the removal of each node in the rainfall network. In each implementation, only one node is removed from the network according to the ranking with replacement (bootstrapping).**

Secondly, we remove successively a larger number of stations, from 1 to 123 stations (10%), considering three cases. In case I, we remove up to the 10% highest ranking stations. This implies that in the first iteration we remove the top-ranked station and in the second iteration we remove the top two stations and so on. Figure 7 shows a clear increase in $\mu$ when more and more influential stations are removed. In case II, up to the 10% lowest ranking stations are successively removed. It can be seen in Fig. 7 that this affects the network efficiency in a positive way: The efficiency increases when the lowest ranking stations are removed. In case III, up to 10% stations are randomly removed. Case III is repeated ten times to understand the effect of random sampling. In general, $\mu$ increases with removing random stations. However, the effect is much lower (in absolute terms) compared to the effect of removing high or low ranking stations, respectively. The variation in $\mu$ between the ten trials and within one trial is caused by randomness. For example, $\mu$ rises instantaneously when the algorithm picks up a high ranking station.

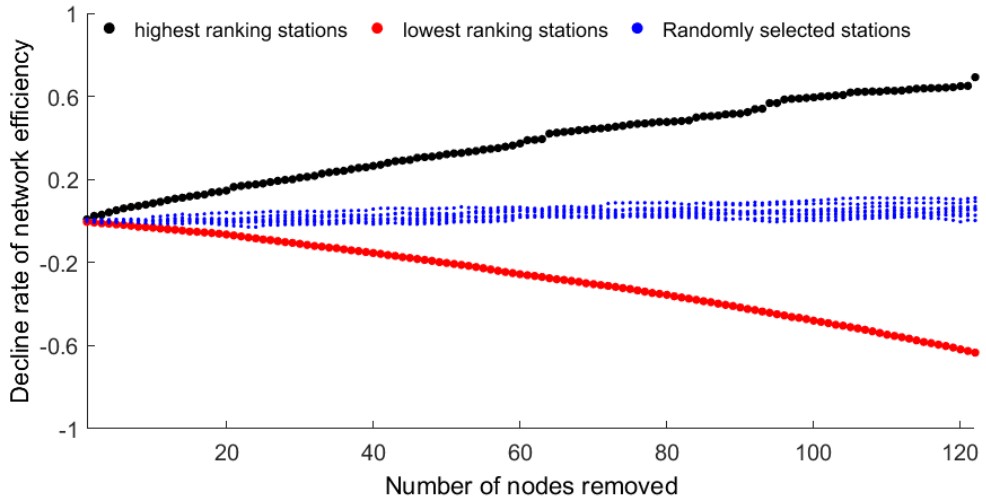

**Figure 7: Decline rate of network efficiency as a function of the number of stations removed from the network. Case I: up to the 10% highest ranking stations are removed (black), case II: up to the 10% lowest ranking stations are removed (red), case III: up to 10% randomly drawn stations are removed (10 trials) (blue).**

## 4.4 Relative Kriging Error

As the second approach to assess the suitability of WDB for identifying influential and expendable stations, we analyse the change in the kriging error when stations are removed from the network. The variogram is kept constant during the network modifications. Similarly to the evaluation using the decline rate of network efficiency in section 4.3, three cases are investigated: removing the 10% highest ranking stations, removing the 10% lowest ranking stations, and ten trials of removing 10% of the stations randomly. The change in the kriging error is calculated for five characteristics, i.e., mean, 90%-, 95%-, 99%-percentile, and number of wet days (Table 1).

Removing the 10% high-ranking stations (case I) leads to positive and high ($\Re > 5\%$) values for all five statistics considered. The kriging error increases substantially when these stations are removed. When the 10% lowest ranking stations (case II) are not considered, the $\Re$ values are small compared to those obtained by removing high ranking stations. The relative errors in estimating the mean, percentile rainfall characteristics (90[th] and 95[th]) and number of wet days at ungauged locations are low (<5%) for the 10% lowest ranking stations, suggesting that these stations do not contribute much information. For two out of five statistics, i.e., mean and number of wet days, removing the 10% lowest ranking stations actually improves the kriging model. Case III, i.e. removing stations randomly, shows mostly positive and high ($\Re > 5\%$) values, because high ranking nodes are removed as well which leads to higher rates of $\Re(\%)$.

**Table 1: Relative kriging error for the three different cases. The relative kriging error for case III is the average across ten trials. Stars indicate a high relative error >5%.**

| Case | Removal of stations | Relative kriging error $\Re$(%) | | | | |
|------|--------------------|------|------|------|------|------|
| | | Mean | 90th percentile | 95th percentile | 99th percentile | Wet days |
| I | 10% highest ranking | 9.3* | 32.9* | 72.3* | 57.1* | 69.1* |
| II | 10% lowest ranking | -2.1 | 4.4 | 3.1 | 11.1* | -1.7 |
| III | 10% randomly selected | 5.4* | 27.3* | 52.3* | 42.6* | 4.1 |

## 5 Discussion

Building on the young science of complex networks, a novel node ranking measure, the weighted degree-betweenness *WDB,* is proposed. It is based on the degree and betweenness centrality measures of the nodes in a network. The comparison of the WDB measure with the existing traditional and contemporary node ranking measures suggests that it is more informative since it is better able to consider the different role of nodes in a complex network. The WDB measure provides a unique value to each node depending on its importance and influence in the network.

Further, this study proposes to use WDB for supporting the optimal design of large hydrometric networks. It is able to rank the nodes in a large network in relation to their importance for the flow of information, mass or energy. This ranking can be used to identify highly influential and expendable hydrometric stations. For example, removing low ranking stations in the German rain gauge network increases the network efficiency considerably, and may even decrease the error of estimating rainfall at ungauged locations. This is explained by the redundancy in the information that those stations provide, which in turn is attributed to the similarity between the gauges due to the common driving mechanisms or spatial similarity as advocated by Tobler's Law of Geography (Tobler, 1970). The results of our analysis suggests that WDB identifies the expendable nodes correctly as shown by the decline rate of efficiency and the insignificant change in relative kriging error. On the other hand, WDB awards stations which provide unique information which cannot be generated from other stations in the network.

We further analyse the characteristics of the stations with the highest ranks. We plot the network (Figure 8a) corresponding to the 10% (~122) high ranking stations, i.e. all the links originating only from these 122 stations. The size and color of each diamond shaped raingauge mark their degree and betweenness. All other stations are plotted in the background without highlighting their degree and betweenness. This sub-network is still difficult to interpret, hence we further plot the connections corresponding to two high ranking stations (Figure 8b) and two low ranking stations (Figure 8c).

Although the degree of these four stations is roughly the same, there is a striking difference in the connections between low and high ranking stations. The connections of low ranking stations are regionally confined, and they rather reflect the similarity in rainfall variability within (homogenous) regions. The plot of high ranking stations in Figure 8a highlights that high rank stations are not limited to high degree or betweenness stations. The latter measures represent the homogeneity (high degree = many similar nodes of similar dynamics) and the path in the network, respectively, whereas WDB represents the connectiveness. This could reflect the critical nodes in pathways of moisture transport, extreme rainfall propagation, or (in case of betweenness) or a handful of stations which are positioned in-between the large communities and unlike most stations they tend to possess intercommunity connections (Halverson and Fleming, 2015; Molkenthin et al., 2015; Tupikina et al., 2016). To test whether the long-range connections of the selected nodes in Figure 8b are a typical feature of high ranking stations, we compute the geographical distance between all the connected raingauges and plot its median (Figure 8d) and 95[th] percentile (Figure 8e) against the node ranking. There is a clear correlation between rank and distance: High ranking stations tend to show longer connections, implicitly affirming that the WDB measure has the potential to capture highly influential nodes in the network.

Further, Fig.8 also in congruence with the results reported by declining rate of Kriging error in section 4.4 and Table 1. Intuitively, "the kriging variance is expected to be greater at a location surrounded by data that are very different from one another (Fig. 8b) than at a location surrounded by similarly valued (Fig. 8c) data" (Goovaerts, 1997; Heuvelink and Pebesma, 2002). And hence, we notice high kriging error (Table 1) in case of influential stations comparative to random selected and low ranking stations. We further noticed that removing the low ranking measures does not have an adverse effect on the estimation of other statistical measures of rainfall thereby verifying the efficacy of the method.

Based on our analysis, we argue that the use of complex networks and the proposed network measure are valuable for the optimal design of hydrometric networks or redesign of existing networks. The proposed node ranking approach differs from the existing approaches as it considers different aspects of the spatio-temporal relationships in observation networks. This measure also has the potential to support the selection of an optimal number of stations for the prediction in ungauged basins (PUBs) and the estimation of missing values by identifying influential stations in the region. For example, the study by Villarini et al., (2008) proposed a simple rule for the number of rain gauges required to estimate areal rainfall with a prescribed accuracy. In such scenarios, WDB measure could be applied to identify prescribed number of influential stations. Further, the proposed method can be applied to gridded satellite data (e.g. rainfall, soil moisture), to locate the strategic points where stations should be installed to ensure a highly efficient observation network.

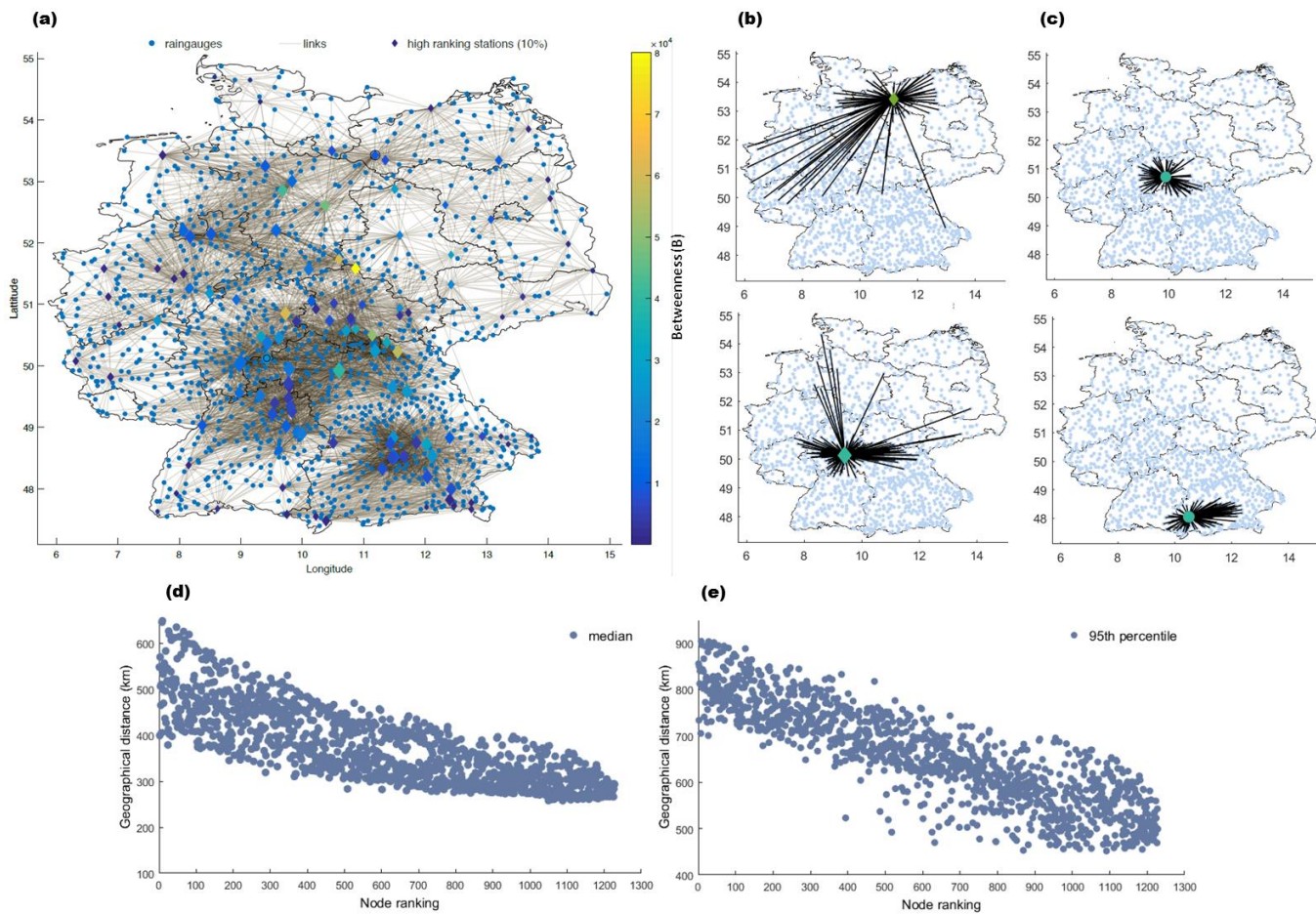

**Figure 8: Connections and location of 10% (~122) highest ranking rain gauges (a). The size and colour of the diamond marker indicate the degree and betweenness of the raingauge. Connections corresponding to two high ranking stations (b, station ID: 21320, 16149) and two low ranking stations (c, station ID: 26132, 20356). Median (d) and 95th percentile (e) geographical distance plotted against node ranking.**

An advantage of the proposed method is its capability to differentiate between the different roles played by individual stations. For example, global bridge nodes are able to control the flow of information, energy or mass between different parts of a network. Hence, they are of highest importance. This capability opens new possibilities for its use in complex networks. For instance, in climate networks an early warning signal could be generated by capturing the flow of information at such points (Donges et al., 2009b; Hlinka et al., 2014).

**6 Conclusions**

This study proposes a novel node ranking measure for identifying the influential and expendable nodes in a complex network. The new network measure weighted degree-betweenness (WDB) combines the existing measures degree and betweenness centrality and considers the neighbourhood of a node. The proposed measure is compared to other measures using synthetic networks. WDB is more sensitive to the different roles of nodes, such as global connecting nodes, hybrid nodes, and local centers, and provides a more informative ranking than the existing node ranking measures.

We propose to use this measure for the optimal design of hydrometric networks. Applying this measure to a network of 1229 rain gauges in Germany allows identifying influential and expendable stations. Two criteria, the decline rate of network efficiency and the kriging error, are used to evaluate the performance of the proposed node ranking measure. The results suggest that the proposed measure is indeed capable of effectively ranking the stations in large hydrometric networks.

Despite the preliminary results of the study, we suggest that the proposed measure is not only useful for optimizing observational networks, but has the potential to support the selection of an optimal number of stations (by determining influential station of the region) to be used in the prediction in ungauged basins, or to support the estimation of missing values, regionalization, and regional flood frequency analysis. When applied to gridded satellite data, it can be used to locate the strategic points where stations should be installed to ensure a highly efficient network. Furthermore, the new network measure has large potentials in other fields where the science of complex networks is used, such as in social networks, infrastructure networks, disease spreading networks, and brain networks.

**Data availability**

The precipitation data was provided by the German Weather Service. The data is publicly accessible at https://opendata.dwd.de/. The data was pre-processed by the Potsdam Institute for Climate Impact Research (Conradt et al., 2012).

**Appendix**

**A. Kriging variogram modelling**

The kriging modelling mandates a theoretical variogram function that is to be fitted with an experimental variogram of the observed data. The experimental variogram ($\gamma(h)$) is calculated from the observed data as a function of the distance of separation (h) and is given by (Adhikary et al., 2015)

$$\gamma(h) = \frac{1}{2N(h)} \sum_{i=1}^{N(h)} [(Y(i) - Y(j))^2] \qquad \text{(A1)}$$

where $N(h)$ is the number of sample data points separated by a distance $h$; , $i$ and j represent sampling locations separated by a distance h; $Y(i)$ and $Y(j)$ indicate values of the observed variable $Y$, measured at the corresponding locations $i$ and

*j* respectively. The theoretical variogram function ($\gamma * (h)$) allows the analytical estimation of variogram values for any distance and provides the unique solution for weights required for kriging interpolation.

The variogram models are a function of three parameters, known as the range, the sill, and the nugget (Fig.A1 (a)). The range is typically the distance where the models first flattens out i.e. station locations separated by distances closer than the range are spatially auto-correlated, whereas locations farther apart than the range are not. The value of $\gamma$ at range is called the sill. The variance of the sample is used as an estimate of the sill. Additionally, many other variogram models have a range parameter that is not equal to a distance where the correlation is zero. Nugget represents measurement error and/or microscale variation at spatial scales that are too fine to detect and is seen as a discontinuity at the origin of the variogram model. The ratio of the nugget to the sill is known as the nugget effect, and may be interpreted as the percentage of variation in the data that is not spatial. The difference between the sill and the nugget is known as the partial sill.

The value of all the parameters and resultant variogram for mean, 90th percentile, 95th percentile, 99th percentile and wet days has been reported in the Table A1 and Figure A1 (b-d) respectively. The variogram has been kept constant during network reductions.

**(a) Typical variogram model**                    **(b) Mean**

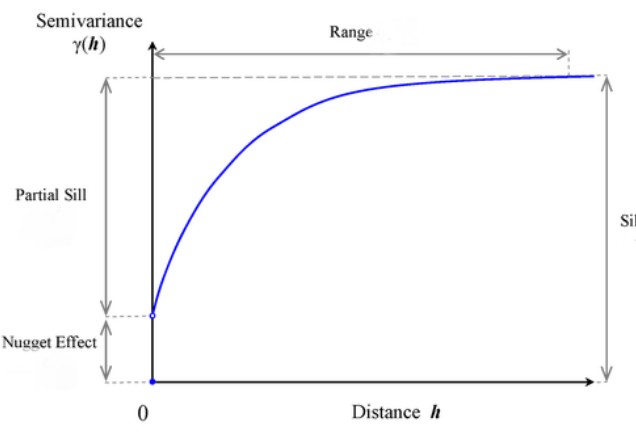
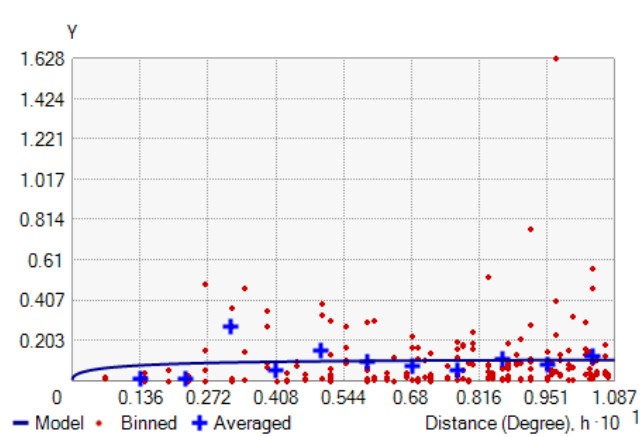

**(c) 90th percentile**                              **(d) 95th percentile**

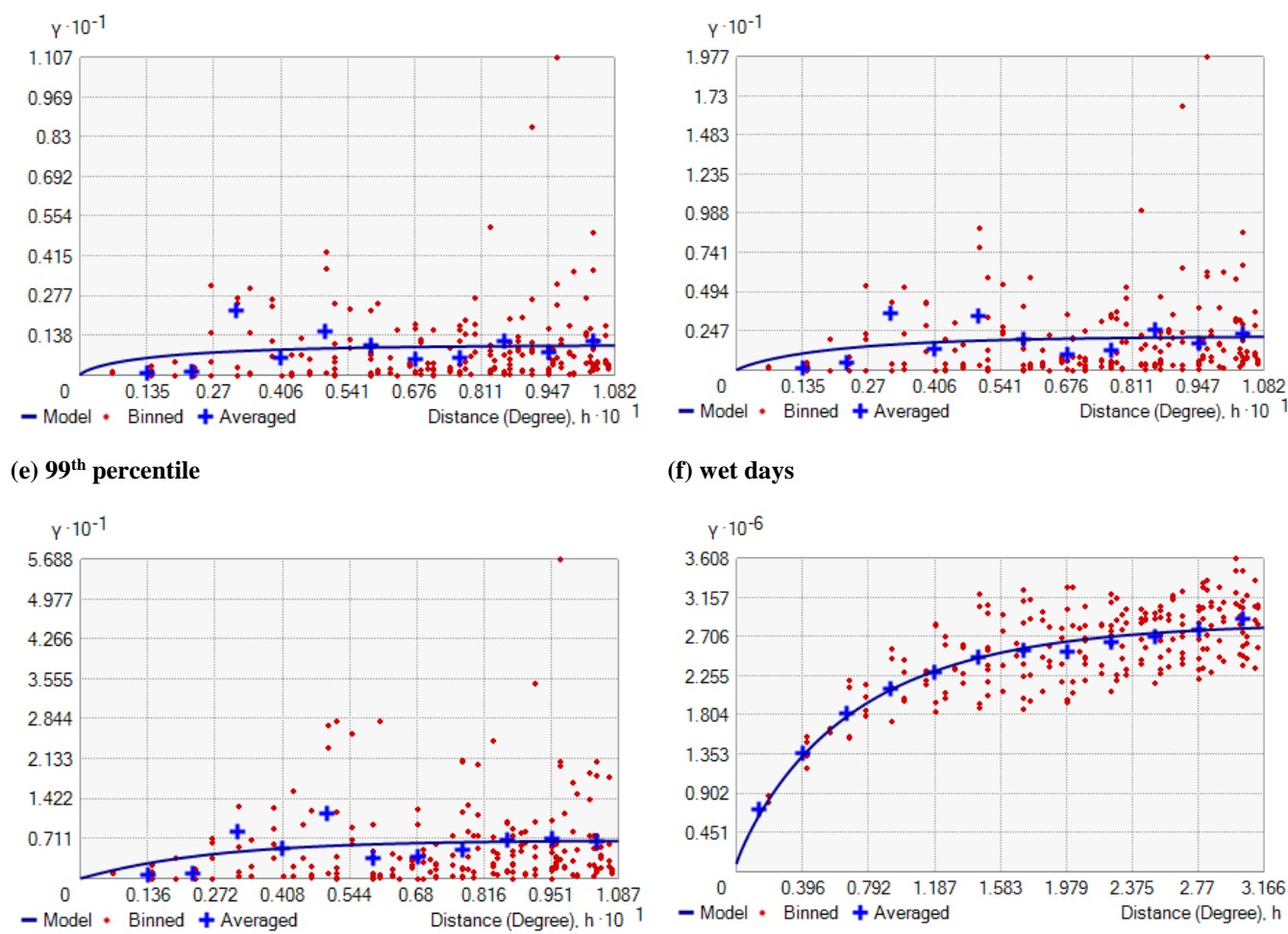

**(e) 99th percentile**

**(f) wet days**

**Figure A1: Typical variogram models (a) and fitted variogram models for mean (b), 90th percentile (c), 95th percentile (d), 99th percentile (e) and wet days (f).**

**Table A1: Parameters values for the fitted variogram.**

| Parameters | Mean | 90th percentile | 95th percentile | 99th percentile | Wet days |
|---|---|---|---|---|---|
| Nugget | 0.0058 | 0 | 0 | 0 | 0.905 |
| Range | 0.0782 | 0.0782 | 0.0782 | 0.0782 | 2.363 |
| Partial sill | 0.103 | 1.055 | 2.140 | 6.808 | 2.771 |

**B. Spatial stationarity check**

Goovaerts (1997, p. 179) states that the theoretical kriging error is dependent on the covariance model and data configuration whereas independent of data values. In a given scenario of constant variogram during network modifications (as mentioned on P17/L7), theoretical kriging error only depends on data configuration (density and distribution of inter-distances between stations). To rule the possibility that these theoretical kriging error also influenced by data values or spatial variance we

double check the spatial stationarity of the measure field of the considered variable. Spatial stationarity means that local variation doesn't change in different areas of the map. For example, 2 data points 5 meters apart in different locations should have similar differences in your measured value. Kriging is not optimal for spatial abrupt changes and break lines. In literature two methods have been proposed to check data's stationarity with a voronoi map symbolizing by entropy (variation between neighbors) or standard deviation and look for randomness.

First check has been performed in the ArcGIS (Geostatistical Analysis →Explore Data →Voronoi Map) on all the considered variables (mean, wet days, 90th-, 95th-, 99th percentile). The corresponding results for entropy voronoi maps show the data set is looking adequately stationary (Fig. B1). However to quantify it, the second check has been performed in the Matlab using run test for randomness with the null hypothesis that the values in the data vector come in random order, against the alternative that they do not. The run test for randomness on all the considered variables rejects the null hypothesis

with a p-value less than .0001. Hence, both methods confirm that kriging model used is spatially stationary, one of the mandatory condition to perform kriging.

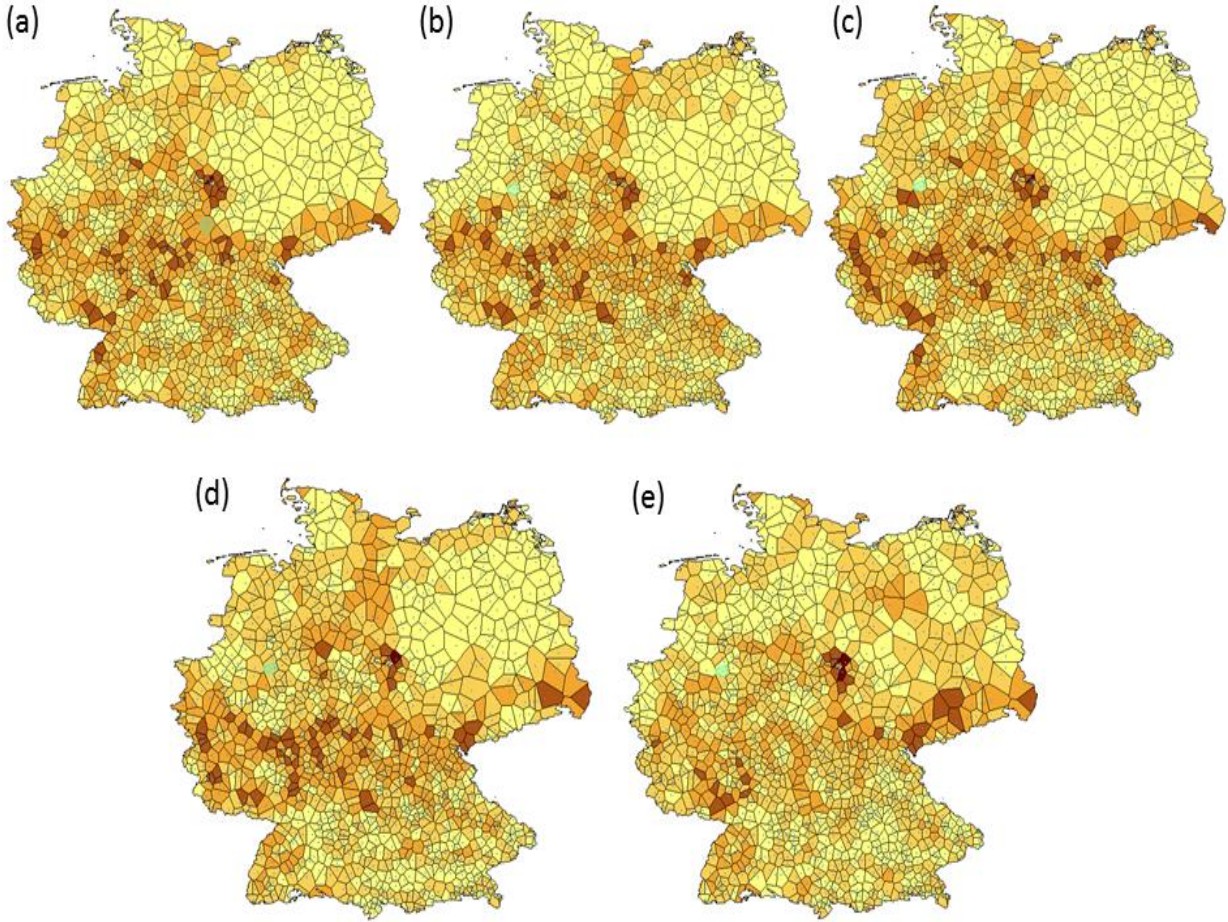

**Figure B1: Representation of mean (a), 90$^{th}$ percentile (b), 95$^{th}$ percentile (c), 99$^{th}$ percentile (d) and wet days's (d) spatial stationarity with a Voronoi map symbolising by entropy (variation between neighbours) using ArcGIS.**

**Competing interests**

5  The authors declare that they have no conflict of interest.

**Acknowledgement**

This research was funded by Deutsche Forschungsgemeinschaft (DFG) (GRK 2043/1) within the graduate research training group Natural risk in a changing world (NatRiskChange) at the University of Potsdam (http://www.uni-potsdam.de/natriskchange). Also, we gratefully acknowledge the provision of precipitation data by the German Weather
10  Service. Ugur Ozturk was partly funded by the Federal Ministry of Education and Research (BMBF) within the project

CLIENT II-CaTeNA (FKZ 03G0878A). The authors gratefully thank the Roopam Shukla (RDII, PIK-Potsdam) for helpful suggestion.

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
