# Peer review of "Optimal Design of Hydrometric Station Networks Based on Complex Network Analysis"

_Hydrology and Earth System Sciences, 2018_

## Referee Comment (RC1) · Anonymous Referee #1 · 4 Apr 2018

The presented work is based on an interesting and attractive idea, namely the transposition of complex network analysis methods to evaluate and support the optimal design of hydrometric networks. A new metrics is proposed to weight and rank the relative importance of the nodes of the network: the weigthed-degree-betweenness (WDB). Two nodes of hydrometric network are considered as connected if the occurrence of heavy events is sufficiently synchronized at the two stations. If I understand correctly stations with strong similarities with other stations will have a large number of connections and hence a high WDB value and conversely. The approach is tested against a large and rich data set composed of 1229 German raingauges. Two criterions are used to compare different strategies to remove 10% of the stations of the network: the so-called network efficiency (average value of the inverse of path lengths between two nodes of

the network) and the interpolation (i.e. kriging ) error. According to the results, removing the lowest ranking stations (stations with the lowest WDB values) has the lowest impact on both criterions, i.e. the proposed ranking measure helps apparently identifying the less influential stations, the station that can be removed from the networks with the most limited consequences on the measurements. This being said, the article appears to draw an extremely counter-intuitive if not absurd conclusion: the stations with the lowest correlation with the other stations of the network, station that a priori provide important additional information, should be removed first. This conclusion is highly questionable and may be explained by the selection of inadequate ranking and efficiency evaluation methods. At least, some further analyses should be conducted before the publication of the manuscript can be considered. The ranking method is selected without considering the final objective and is probably inadequate. An explanation is clearly missing at the beginning of the manuscript to explain why the network construction method and the proposed WDB are suited to rate the relative information content of the stations of the network. My feeling is that the proposed approach leads to attribute the highest ranks to the stations with the lowest relative information content which is exactly the opposite of what is meant. Moreover, the validation based on kriging necessitates a more in-depth analysis and probably further tests to be conducted. The authors consider a so-called kriging error which definition has first to be clarified. It seems to be a theoretical kriging error standard deviation provided by the ArcGIS software geostatistical extension. In fact, this standard deviation depends on the location. What is provided is certainly an average value over the whole considered area – this of course has to be clarified by the authors. This standard deviation depends on the network density, on the variance of the rainfall fields and on the characteristics of the variogram. At least the variogram and variance of fields have to be provided as a support to the analysis for the various tested networks. Removing atypical rain gauges can easily have tricky impact on the average theoretical kriging error standard deviation: the lower density of the network may be partly compensated by higher homogeneity of the measured rainfall fields (higher decorrelation distances and lower field variances).

This compensation effect could explain the modest influence or even the positive effect in table 4 for case 2. In fact, the theoretical error standard deviations are two much dependent on the network itself to enable comparisons between network structures. More classical comparison methods, based for instance on observed interpolation errors, should absolutely be selected and tested by the authors. The distance between interpolated fields obtained with the complete (reference) and reduced networks could for instance be evaluated. Interpolation errors could also be computed based on a leave-one-out sampling method providing more realistic estimates of real interpolation errors. Of course the leave-one-out test station should be selected before the network reduction methods are applied. These verification methods are computationally probably expensive but absolutely necessary. According to these doubts concerning the adequacy of the proposed method and the soundness of the conclusions, I do not recommend the publication of the manuscript and the real-world application of the suggested ranking method unless the proposed additional verifications are conducted.

---

## Author Comment (AC1) · 19 Apr 2018

We thank the reviewer for investing his/her valuable time in our manuscript. We think that the reviewer has misunderstood some of the material, and we apologize if our presentation has not been totally clear in all instances. We understand that conciseness is particularly important for manuscripts like this which builds on emerging ideas in the very fast-evolving field of complex network theory, as well as on new ideas around similarity measures, such as event synchronization, which are rather new in hydrology.

We have responded (in black) to each reviewer comment (in red).

**General comments**

The presented work is based on an interesting and attractive idea, namely the transposition of complex network analysis methods to evaluate and support the optimal design of hydrometric networks. A new metrics is proposed to weight and rank the relative importance of the nodes of the network: the weighted-degree-betweenness (WDB). Two nodes of the hydrometric network are considered as connected if the occurrence of heavy events is sufficiently synchronized at the two stations. ==If I understand, correctly stations with strong similarities with other stations will have a large number of connections and hence a high WDB value and conversely.== The approach is tested against a large and rich data set composed of 1229 German raingauges. Two criterions are used to compare different strategies to remove 10% of the stations of the network: the so-called network efficiency (average value of the inverse of path lengths between two nodes of the network) and the interpolation (i.e., kriging) error. According to the results, removing the lowest ranking stations (stations with the lowest WDB values) has the lowest impact on both criterions, i.e., the proposed ranking measure helps apparently identifying the less influential stations, the station that can be removed from the networks with the most limited consequences on the measurements.

We thank the reviewer for a constructive summary of our manuscript and also for his/her critical and supportive suggestions. Your feedback is vitally important to increase the readability of the work.

**Major comments**

If I understand correctly stations with strong similarities with other stations will have a large number of connections and hence a high WDB value and conversely.

This is a misunderstanding. The ranking measure that we propose (WDB) is not simply related to the number of links. In fact, the issue that is raised by the reviewer is one of the limitations of the traditional node ranking measure called degree: high number of links → high degree → higher rank. This limitations was one of the reasons why we developed the new measure WBD.

The difference between degree and our new measure WBD is discussed through the artificial network example in Fig. 2 and Table 2. For instance, node 5 has the highest WDB score but the lowest decree score. This means that the most important node (according to our measure) has the lowest number of links in the network!

The difference between degree and WBD and the limitations of the traditional measure decree are also explained in the text, e.g. on page 4, line 9 ("...*The degree can explain the importance of nodes to some extent, but nodes that own the same degree may not play the same role in a network. For instance, a bridging node connecting two important nodes might be very relevant though its degree could be much lower than the value of less important nodes...*") or on page 6, line 7 ("*... Degree is limited as node ranking measure since it cannot distinguish between different roles in the network. For example, nodes 5, 7, and 8 have the same degree ($k_i=2$), but node 5 serves as bridge node linking the two parts of the network...*").

However, to make it even more explicit, we propose to add the following statement (marked in yellow) in section 2.2: "*The degree k of a node in a network counts the number of connections linked to the node directly. For example, the degree of nodes 1, 2 and 4 in network N1 (Fig. 1a) is 1 and for node 3 is 3. In the network N2 (Fig. 1b),*

*all nodes have degree 3. The degree can explain the importance of nodes to some extent, but nodes that own the same degree may not play the same role in a network. For instance, a bridging node connecting two important nodes might be very relevant though its degree could be much lower than the value of less important nodes. Moreover, stations with strong similarities with other stations will have a large number of connections resulting in a high degree and hence a high rank and conversely. "*

This being said, the article appears to draw an extremely counter-intuitive if not absurd conclusion: the stations with the lowest correlation with the other stations of the network, station that a priori provide important additional information, should be removed first. This conclusion is highly questionable and may be explained by the selection of inadequate ranking and efficiency evaluation methods. **At least, some further analyses should be conducted before the publication of the manuscript can be considered.** The ranking method is selected without considering the final objective and is probably inadequate.
My feeling is that the proposed approach leads to attribute the highest ranks to the stations with the lowest relative information content which is exactly the opposite of what is meant.

This statement of the reviewer seems to be a consequence of his/her misunderstanding of the rank measure. We have already mentioned that our proposed measure (WDB) identifies stations which provide additional information, for example, on page 14, line 15: *"... it awards stations which provide unique information which cannot be generated from other stations in the network ..."* However, to be absolutely clear, we will add a discussion on how our measure differs from a correlation analysis and accounts for the information content of stations (see response to next reviewer comment).

There is one situation where our method would require additional care: Let's imagine a node that is unrelated to other nodes (no links). Physically, one might imagine this scenario in a meteorological sub-region characterized by fine-scale convective thunderstorms with sparse rain gauge coverage. Hence, precipitation event synchronization across rain gauges in that sub-region would be poor. In that case, indeed, this station would not be the part of constructed network, and would not be ranked. This station should be treated carefully as it provides unique information. We will add a remark on this particular situation in the revision.

An explanation is clearly missing at the beginning of the manuscript to explain why the network construction method and the proposed WDB are suited to rate the relative information content of the stations of the network.

We thank the reviewer for this proposal and propose to add two paragraphs in the introduction:

"… The study aims to identify influential and redundant stations based on the relative information content. In the past several measures, such as zero-lag correlation or time-delayed correlation, have been used. However, these measures are limited by the underlying assumptions, e.g. measuring linear relations. Further, they give equal weight to high and low rainfall values, whereas the main information content in a rainfall time series is embedded in the larger values. In this study, we use event synchronization (ES) as a similarity measure for the network construction. ES is a suitable measure for event like, non-Gaussian data such as precipitation (Stolbova et al., 2014; Tass et al., 1998). It has advantages over other time-delayed correlation techniques (e.g., Pearson lag correlation), as it allows us to define the event time series by determining the threshold, and as it uses a dynamic time delay. The latter refers to a time delay that is adjusted according to the two time series being compared, which allows for better adaptability to the region of interest…."

"… Identifying key nodes in complex networks has attracted increasing attention in recent years (Chen et al., 2012; Hou et al., 2012; Jensen et al., 2016; Kitsak et al., 2010; Zhang et al., 2013). There are several methods to evaluate the importance of nodes (Hu et al., 2013). Degree (k), betweenness centrality (B) and closeness centrality (CC) are the methods commonly used in complex networks (Gao et al., 2013). Studies in different disciplines have shown that degree and betweenness centrality often outperform other node-ranking measures (Gao et al., 2013; Liu et al., 2016). In this study, we propose a novel measure called weighted degree-betweenness (WDB), combining degree *(k)* and betweenness centrality *(B)*, which combines the advantages of both. Our case studies show that the proposed measure WDB has an even higher discrimination power compared to betweenness centrality and that it effectively ranks the nodes in the network. WDB is more sensitive to the different roles of nodes, such as global connecting nodes, hybrid nodes, and local centers, and provides a more informative ranking than the existing node ranking measures.

Moreover, the validation based on kriging necessitates a more in-depth analysis and probably further tests to be conducted. The authors consider a so-called kriging error which definition has first to be clarified. It seems to be a theoretical kriging error standard deviation provided by the ArcGIS software geostatistical extension. In fact, this standard deviation depends on the location. What is provided is certainly an average value over the whole considered area – this, of course, has to be clarified by the authors. This standard deviation depends on the network density, on the variance of the rainfall fields and on the characteristics of the variogram. At least the variogram and variance of fields have to be provided as a support to the analysis for the various tested networks.

We thank the reviewer for highlighting the need for such more detailed information. We use the kriging standard error (KSE) which is the square root of the kriging variance (Adhikary et al., 2015; Xu et al., 2018). We estimate the kriging standard error across the entire study area. We will add the following information in the revision:

*The kriging variance $\sigma_z^2(x_o)$, in the ordinary kriging can be computed as*

$$\sigma_z^2 = \mu_z + \sum_{i=1}^{n} w_i \gamma(h_{oi}) \quad for \sum_{i=1}^{n} w_i = 1$$

*where $\gamma(h)$ is the variogram value for the distance h; $h_{0i}$ is the distance between observed data points xi and xj; $\mu z$ is the Lagrangian multiplier in the Z scale; $h_{0j}$ is the distance between the unsampled location $x_0$ (where estimation is desired) and sample locations $x_i$; and n is the number of sample locations.*
*The square root of the kriging variance, also named as kriging standard error (KSE), is used as a gauge network evaluation factor which can reflect the performance of optimal gauge combination".*
In addition, we will add the variance of the rainfall fields and the variogram, in the revised version.

Removing typical rain gauges can easily have a tricky impact on the average theoretical kriging error standard deviation: the lower density of the network may be partly compensated by the higher homogeneity of the measured rainfall fields (higher decorrelation distances and lower field variances). This compensation effect could explain the modest influence or even the positive effect in table 4 for case 2. In fact, the theoretical error standard deviations are two much dependent on the network itself to enable comparisons between network structures.

We thank the reviewer for his/her suggestion. However, we would like to bring to your attention that the main motivation was to see, how removing the low ranking and high ranking stations impacts the kriging error variance and thereby verifying the efficiency of the method. We would like to point out that the proposed WDB method is independent of the distances between the nodes and considers only the similarity. We agree that when there is strong homogeneity in the rainfall field then a smaller number of stations is required to capture the variance. We believe that 3 different networks, whereas each network is reduced by 10% of its stations, can be compared to the full network by the kriging approach, as for example done in Adhikary et al., 2015; Kassim and Kottegoda, 1991; Xu et al., 2018; Yeh et al., 2006. However, to better analyze the effect of the proposed node ranking in reducing a station network, we will perform additional analyses for inclusion in the revised version.

More classical comparison methods, based for instance on observed interpolation errors, should absolutely be selected and tested by the authors. The distance between interpolated fields obtained with the complete (reference) and reduced networks could, for instance, be evaluated. Interpolation errors could also be computed based on a leave-one-out sampling method providing more realistic estimates of real interpolation errors. Of course, the leave-one-out test station should be selected before the network reduction methods are applied. These verification methods are computationally probably expensive but absolutely necessary.

We will perform additional comparisons by analyzing the interpolation errors, based on a leave-one-out sampling method.

According to these doubts concerning the adequacy of the proposed method and the soundness of the conclusions, I do not recommend the publication of the manuscript and the real-world application of the suggested ranking method unless the proposed additional verifications are conducted.

Again, we would like to thank the reviewer for his/her comments. The doubts raised are on the one hand a consequence of a misunderstanding by the reviewer. This confusion between the two measures (degree and WDB) can be easily rectified by being even more explicit in the description of the method. On the other hand, we feel that the reviewer raised important issues about the comparison using the kriging method. We will provide further details about the kriging and we will add further comparison methods to better evaluate our method.

**Reference**

Adhikary, S. K., Yilmaz, A. G. and Muttil, N.: Optimal design of rain gauge network in the Middle Yarra River catchment, Australia:, Hydrol. Process., 29(11), 2582–2599, doi:10.1002/hyp.10389, 2015.

Chen, D., Lü, L., Shang, M.-S., Zhang, Y.-C. and Zhou, T.: Identifying influential nodes in complex networks, Phys. Stat. Mech. Its Appl., 391(4), 1777–1787, doi:10.1016/j.physa.2011.09.017, 2012.

Gao, C., Wei, D., Hu, Y., Mahadevan, S. and Deng, Y.: A modified evidential methodology of identifying influential nodes in weighted networks, Phys. Stat. Mech. Its Appl., 392(21), 5490–5500, doi:10.1016/j.physa.2013.06.059, 2013.

Hou, B., Yao, Y. and Liao, D.: Identifying all-around nodes for spreading dynamics in complex networks, Phys. Stat. Mech. Its Appl., 391(15), 4012–4017, doi:10.1016/j.physa.2012.02.033, 2012.

Jensen, P., Morini, M., Karsai, M., Venturini, T., Vespignani, A., Jacomy, M., Cointet, J.-P., Mercklé, P. and Fleury, E.: Detecting global bridges in networks, J. Complex Netw., 4(3), 319–329, doi:10.1093/comnet/cnv022, 2016.

Kassim, A. H. M. and Kottegoda, N. T.: Rainfall network design through comparative kriging methods, Hydrol. Sci. J., 36(3), 223–240, doi:10.1080/02626669109492505, 1991.

Kitsak, M., Gallos, L. K., Havlin, S., Liljeros, F., Muchnik, L., Stanley, H. E. and Makse, H. A.: Identification of influential spreaders in complex networks, Nat. Phys., 6(11), 888–893, doi:10.1038/nphys1746, 2010.

Liu, J., Xiong, Q., Shi, W., Shi, X. and Wang, K.: Evaluating the importance of nodes in complex networks, Phys. Stat. Mech. Its Appl., 452, 209–219, doi:10.1016/j.physa.2016.02.049, 2016.

Stolbova, V., Martin, P., Bookhagen, B., Marwan, N. and Kurths, J.: Topology and seasonal evolution of the network of extreme precipitation over the Indian subcontinent and Sri Lanka, Nonlinear Process. Geophys., 21(4), 901–917, doi:10.5194/npg-21-901-2014, 2014.

Tass, P., Rosenblum, M. G., Weule, J., Kurths, J., Pikovsky, A., Volkmann, J., Schnitzler, A. and Freund, H.-J.: Detection of n : m Phase Locking from Noisy Data: Application to Magnetoencephalography, Phys. Rev. Lett., 81(15), 3291–3294, doi:10.1103/PhysRevLett.81.3291, 1998.

Xu, P., Wang, D., Singh, V. P., Wang, Y., Wu, J., Wang, L., Zou, X., Liu, J., Zou, Y. and He, R.: A kriging and entropy-based approach to raingauge network design, Environ. Res., 161, 61–75, doi:10.1016/j.envres.2017.10.038, 2018.

Yeh, M.-S., Lin, Y.-P. and Chang, L.-C.: Designing an optimal multivariate geostatistical groundwater quality monitoring network using factorial kriging and genetic algorithms, Environ. Geol., 50(1), 101–121, doi:10.1007/s00254-006-0190-8, 2006.

Zhang, X., Zhu, J., Wang, Q. and Zhao, H.: Identifying influential nodes in complex networks with community structure, Knowl.-Based Syst., 42, 74–84, doi:10.1016/j.knosys.2013.01.017, 2013.

---

## Referee Comment (RC2) · Anonymous Referee #1 · 20 Apr 2018

I am not convinced that Fig.2 and Table 2 really help understand what the result of the implementation of the proposed ranking method gives if applied to a real observation network. The example in Fig.2 is extremely simple and can hardly be extrapolated to a raingauge network including hundreds of gauges. It is unclear what node 5 in figure 2, which illustrates the added value of the proposed ranking method, really stands for. What about raingauges that are poorly correlated with other gauges that could rather appear as dead-ends or even isolated nodes in the build network ? Some extracted maps from figure 4 showing on a limited size area, the topography along with the location and resulting ranks of the raingauges and maybe also the location of the 10% higher ranked removed gauges could improve a lot the presentation of the method.

Moreover, I confirm that additional validation is needed and this is acknowledged by

the authors. At least the resulting variance of the rainfall fields corresponding to the various tested networks in table 4 should be provided. If significantly modified by the gauge selection method – probably moderately for the random selection method – it could have a major impact on the figures in table 4. This should be considered in the interpretations. But evaluations based on a leave-one-out approach should also be conducted.

---

## Short Comment (SC1) · 31 May 2018

The paper is novelty and innovative and in my opinion it should be published. I believe that the 'complex network paradigm' offers a very useful reference framework that can stimulate the development of new approaches and methods to solve relevant issues in the field of hydrology. Therefore, the exploration of new methods within such framework, as those proposed in the present manuscript, should be encouraged. Despite this, I think that the quality of the paper should be improved in some points, because in the present form could be misunderstood, especially from people who are more familiar with the traditional hydrological approaches.

Mathematical critical points:

[Figure]

A) I think that there is an error in the formula (3) since I was not able to calculate the values of WDB, for the simple networks of figure 2, reported in table 2. In each case an effort to provide an hydrological meaning of such metric could help the readers to better evaluate the worth of the proposal. B) Betweenness in figure 2, coherently with the definition given in paragraph 2.2, it should be $4(12) - 5(12) - 6(10)$ instead of $4(24)$ $5(24)$ $6(20)$. C) In paragraph 4.2, in order to select the most synchronized nodes of the network, authors assume the 95th percentile of the values of adjacency matrix elements. May authors provide a justification of such assumption? Just a suggestion: what about if you use all that edges that are significant? An edge is significant if the synchronization value exceeds the 95th percentile of the synchronization obtained by two synthetic variables that have the same number of events positioned randomly in the time series. Authors can run synchronization on 100 random time series (couples) and you get the 95th percentile of synchronization for these synthetic values. D) In the appendix authors use an event synchronization formula which is a modified version of Quiroga et al 2002. However this formulation is still wrong (Q can exceed 1, as stated in Conticello et al 2018. This is due to double count of the events.). If authors use a dynamical tau, you should use the formulation of Kreuz et al 2015, 2016 (Spiky), if you use a defined tau, you should use Conticello et al 2018. (In this case it seems it is dynamical, so Kreuz will be fine).

Physical Interpretation critical point:

D) I have a doubt: what about single nodes which are not synchronized with others, but due to a particular location or specific hydrological conditions have a particular informative content in representing a spatial or temporal variability of the region considered? Maybe, in the case of a very dense network this is not the case, but this issue should be useful to clarify. Let's make a simple example: look at the figure below (attached). You have 10 nodes. The maximum number of undirected edges you can have is $\frac{1}{2}$*(10x10)-10 = 40 (1/2 because it's symmetric, 10x10 is n_nodes x n_nodes, -10 because you don't want to get self synch). In this case, if you want to retain those edges that exceed

95th percentile you get only 2 edges. With this configuration you will drop a lot of stations. Maybe I misunderstood and you want to retain the 95th percentile of all the 100 edges (you should explain clearly in the paper in order to make it reproducible). In any case, following your optimization approach, your network improves when you drop 1-2-3-8 nodes. Your efficiency grows because the area that you are describing it is getting smaller. With this approach you lose all the unique information coming from parts of the areas that need more station. So, from my point of view, maybe the stations that you should drop are 4-6-9-10 because they are more synchronized with 5 and 7, and you need only these stations to represent these subnetworks. I think the betweennes (and WDB) should be used to synthesize data and reduce the uncertainty due to the curse of dimensionality.

Description and Style critical points:

E) in the introduction there is just a list of the classical approaches used for the design of hydrometric networks. These approaches have the goal to identify the optimum number and locations of measurement points able to allow the most reliable possible representation of the spatial and temporal variability of the hydrological variable observed. Since a completely new method is proposed, and not a simple improving of existing ones, some justifications could be provided: a) why is it necessary, or opportune, to explore new approaches offered by the application of the complex network paradigm ; b) what are the potentiality of such more general approach to hydrology. In the specific case faced in the manuscript, some issues about the limitations or drawbacks of the classical approaches could be better underlined, as for instance: they are too much complex, empirical, based on not fully consistent hypothesis, etc, etc. Furthermore, is it possible to emphasize that the proposed approach, based on the complex network paradigm, is potentially more promising than existing ones? F) the metrics of the complex network reported in table 1 - degree, betweeness, etc - are not clearly explained to allow people who are not familiar with complex network, to understand well their meaning. G) In paragraph 3.3.1 the Decline Rate of Network Efficiency

is introduced. I had some difficulties to understand the meaning of such variable. N is not defined. I try to read the cited paper ( Liu 2016) but it use exactly the same words, and this paper in turn cites a paper in Chinese language. How do you measure dij ? i.e. is it the shortest path between nodes ni and nj equal to the sum of the links between ni and nj or it is the number of the shortest paths? Can any hydrological meaning be attributed to such measurement of network efficiency. H) A figure about the network of the German hydrometric stations, obtained by the application of the method, could help to understand how the proposed method works. I figure out that likely more independent networks should arise from the application of such approach because the 95th percentile threshold of the values of adjacency matrix elements .

---

## Referee Comment (RC3) · Anonymous Referee #2 · 26 Jun 2018

This manuscript introduced the use of complex network analyses for designing optimal hydrometric networks. I find the concept interesting, but the presentation is still not ready for publication. Overall, I think a major revision is necessary.

I find the concept interesting, but the authors somewhat fail to explain what the advantage of this method is, and to make me really understand what the network analyses will mean in the case of hydrometeorological observations. It is clear how a linear network can be defined, as in Figure 1, but I find it difficult to imagine the network that is built from the event synchronization. Maybe the authors could show a small example where only a few (imaginary) stations are analysed with the network methodology. Then it can be shown how and why some stations are redundant and can be removed. The real case example from Germany is interesting, but with such a high number of

stations, it is challenging to understand what actually happens.

I am not convinced by the use of a somewhat subjective cutoff value for the Qs to define the network, without at least a much deeper discussion around the effect. This will to a large degree ignore the level of similarity, it is just a yes/no transformation. Increasing or decreasing the threshold could drastically change the importance of the nodes in the network. Two stations with similarity just above the threshold will be treated the same way as two stations which are almost identical. On the other hand, two stations just under the threshold are treated completely different than the stations just above, even if their similarities are almost the same.

The authors do several times mention the importance of global bridge nodes, and the possibilities these give in analyses of complex networks. For example: "For instance, in climate networks and early warning signal could be generated by capturing the flow of information at such points." This might be explained better in some of the references, but it should anyway be better explained what a global bridge node really means in hydrometeorological network, and what kind of information flow we could particularly capture from this node.

I noticed that also the other reviewers asked for some improvements regarding the relative kriging errors. In addition to what they wrote, I was not sure whether the variogram is recomputed when stations are removed. If this is done, then variogram fitting is a science in itself, whether done manually or automatically, and this can lead to changes in the kriging error, making small changes more a result of random changes. The kriging error should normally not decrease when you remove stations, so the reduction in table 4 for the mean is most likely because the variogram has been fitted differently. When kriging error is used to estimate network modifications, the variogram is therefore usually kept constant, to avoid having to also analyse the variogram fitting. The larger changes are still significant.

Some smaller issues:

P2L25 the sentence is somewhat contradictory to the previous one, try to rephrase.

P13 - Fig 6 Remove 10% from the x-label

P16 Eq. A3 explain why the numbers are 1 and $\frac{1}{2}$ in the equation.

---

## Author Comment (AC2) · 9 Aug 2018

Anonymous Referee #1 We thank the reviewer for investing his/her valuable time in our manuscript. Your feedback is vitally important to increase the readability of the work. We have responded (in black) to each reviewer comment (in red).

Best wishes,

On behalf of all co-authors

Please also note the supplement to this comment:

[Figure]

https://www.hydrol-earth-syst-sci-discuss.net/hess-2018-113/hess-2018-113-AC2-supplement.pdf

[Figure]

**Supplement:**

We thank the reviewer for investing his/her valuable time in our manuscript. Your feedback is vitally important to increase the readability of the work.
We have responded (in black) to each reviewer comment (in red).

**General comments**

I am not convinced that Fig.2 and Table 2 really help to understand what the result of the implementation of the proposed ranking method gives if applied to a real observation network. The example in Fig.2 is extremely simple and can hardly be extrapolated to a raingauge network including hundreds of gauges. It is unclear what node 5 in figure 2, which illustrates the added value of the proposed ranking method, really stands for.
We agree with the reviewer that from the discussion of this simple example, one cannot derive the conclusion that our method also works for a raingauge network with hundreds of stations. However, we do not derive this conclusion from this example, but use it to motivate the development of the new node ranking measure. Identifying nodes with interesting positions in a network is useful to extract meaningful information from large datasets. Over the year numerous measures have been introduced for this purpose, such as degree and betweenness centrality. In this example we demonstrate the deficits of the existing measures. We feel that this demonstration is better done with a very simple example where the reader can easily understand and reproduce the calculations. Independently from this example, the proposition that our new measure is useful for large networks of raingauges is discussed and evaluated using the decline rate of network efficiency and the kriging error (section 4.3, 4.4). In the revision, we will make (even more) clear that this example is used for easy understanding and motivation, and that we do not derive that our new measure works for the large raingauge network as well.

The role of node 5 in Fig. 2 can be understood with the followings simple theoretical example.

In climate networks, local centers correspond to nodes which are important for local climate phenomena, while bridges correspond to nodes which connect different subsystems of climate, such as monsoon and El-Nino, leading to teleconnections (Paluš, 2018). Bridge node spread a process to the entire network with more force than a local center in a community, even when the local center has more spreading power (more connection) than the bridging node (Lawyer, 2015). By deducing the globe bridge node in a spatial hydrometeorological network, most of the process flow information can be captured.

What about raingauges that are poorly correlated with other gauges that could rather appear as dead-ends or even isolated nodes in the build network?
We interpret this question as a modified question of the reviewer in his/her first review: *"If I understand correctly stations with strong similarities with other stations will have a large number of connections and hence a high WDB value and conversely".* As we mentioned in our previous response letter the ranking measure that we propose (WDB) is not simply related to the number of links. Hence, the ranking of the nodes will depend on their information contribution to the network, see for example, on page 14, line 15: *"... it awards stations which provide unique information which cannot be generated from other stations in the network ...".* However, we agree with the reviewer that such a situation needs to be considered carefully. We propose to add a discussion on such situations in the revision, as we have already indicated in our first response to reviewer

**1 (… There is one situation where our method would require additional care: Let's imagine a node that is unrelated to other nodes (no links). Physically, one might imagine this scenario in a meteorological sub-region characterized by fine-scale convective thunderstorms with sparse rain gauge coverage. Hence, precipitation event synchronization across rain gauges in that sub-region would be poor. In that case, indeed, this station would not be the part of the constructed network, and would not be ranked. This station should be treated carefully as it provides unique information. …)**

Some extracted maps from figure 4 showing on a limited size area, the topography along with the location and resulting ranks of the raingauges and maybe also the location of the 10% higher ranked removed gauges could improve a lot the presentation of the method.

We thank the reviewer for her/his excellent suggestion. In the revised version we will attempt to incorporate this suggestion.

Moreover, I confirm that additional validation is needed and this is acknowledged by the authors. At least the resulting variance of the rainfall fields corresponding to the various tested networks in table 4 should be provided. If significantly modified by the gauge selection method – probably moderately for the random selection method – it could have a major impact on the figures in table 4. This should be considered in the interpretations. But evaluations based on a leave-one-out approach should also be conducted.

As recommended by the reviewer in her/his review letter we assure to add a new validation scheme based on leave-one-out approach in the revised version. We will also include variance and variogram of various tested networks.

---

## Author Comment (AC3) · 15 Aug 2018

We thank the reviewer for investing his/her valuable time in our manuscript. We understand that conciseness is particularly important for manuscripts like this which builds on emerging ideas in the very fast-evolving field of complex network theory, as well as on new ideas around similarity measures, such as event synchronization, which is rather new in hydrology. We have responded (in black) to each reviewer comment (in red).

Please also note the supplement to this comment:
https://www.hydrol-earth-syst-sci-discuss.net/hess-2018-113/hess-2018-113-AC3-
supplement.pdf

─────────────────────────
113, 2018.

**Supplement:**

We thank the reviewer for investing his/her valuable time in our manuscript. We understand that conciseness is particularly important for manuscripts like this which builds on emerging ideas in the very fast-evolving field of complex network theory, as well as on new ideas around similarity measures, such as event synchronization, which is rather new in hydrology.

We have responded (in black) to each reviewer comment (in red).

General comments

This manuscript introduced the use of complex network analyses for designing optimal hydrometric networks. I find the concept interesting, but the authors somewhat fail to explain what the advantage of this method is, and to make me really understand what the network analyses will mean in the case of hydrometeorological observations. *It is clear how a linear network can be defined, as in Figure 1, but I find it difficult to imagine the network that is built from the event synchronization.*

We thank the reviewer for a constructive summary of our manuscript and also for his/her critical and supportive suggestions. Your feedback is vitally important to increase the readability of the work.

We agree with the reviewer that network construction using ES is not that trivial, since complex networks and event synchronization have hardly been used in hydrology. Hence, we propose to insert the following schematic figure with modifications in a revised version to better explain the network construction using event synchronization. All the equations and symbols has been explained in the main text of manuscript.

**Network construction**

[Figure]

*Step 1.* Apply a threshold to time series of each grid point (*i* and *j*) to obtain extreme event series

[Figure]

*Step 2.* Event synchronization – use time lags to compare individual events between two grid points

[Figure]

*Step 3.* Construct the network by creating links between points with the highest synchronization values

$$A_{ij} = \begin{cases} 1, & if \quad Q_{ij} > \theta_{ij}^{Q} \\ 0, & else. \end{cases}$$

$Q_{ij}$ – is a correlation matrix

$\theta_{ij}^{Q}$ – is a threshold

$A_{ij}$ – is an adjacency matrix

[Figure]

Figure 1: Schematic of network construction using event synchronization (ES). All the equations and symbols has been explained in the main text.

Major comments

Additional synthetic case study for expandable stations: Maybe the authors could show a small example where only a few (imaginary) stations are analyzed with the network methodology. *Then it can be shown how and why some stations are redundant and can be removed.* The real case example from Germany is interesting, but with such a high number of stations, it is challenging to understand what actually happens.

The specific application to use the WDB measure for ranking raingauges in Germany may indeed be difficult to understand. Reviewer #1 (in RC2) suggests that "... Some extracted maps from figure 4 showing on a limited size area, the topography along with the location and resulting ranks of the raingauges and maybe also the location of the 10% higher ranked removed gauges could improve a lot the presentation of the method. ..." In the revised version we will attempt to incorporate this suggestion. We think that this suggestion helps understanding in detail what actually happens.

Threshold cutoff justification: I am not convinced by the use of a somewhat subjective cutoff value for the Qs to define the network, without at least a much deeper discussion around the effect. This will to a large degree ignore the level of similarity, it is just a yes/no transformation. Increasing or decreasing the threshold could drastically change the importance of the nodes in the network. Two stations with similarity just above the threshold will be treated the same way as two stations which are almost identical. On the other hand, two stations just under the threshold are treated completely different than the stations just above, even if their similarities are almost the same.

We thank the reviewer for raising the concern with the subjectivity of threshold. In the revised version we will provide a sensitivity analysis to quantify the effect of the cutoff values.

Global bridge node: The authors do several times mention the importance of global bridge nodes, and the possibilities these give in analyses of complex networks. For example*: "For instance, in climate networks an early warning signal could be generated by capturing the flow of information at such points."* This might be explained better in some of the references, but it should anyway be better explained what a local center and global bridge node really means in the climate network, and what kind of information we could particularly capture from this node.

In climate networks, local centers correspond to nodes which are important for local climate phenomena, while bridges correspond to nodes which connect different subsystems of climate (Jensen et al., 2016), such as the Asian monsoon and El Niño/Southern Oscillation, leading to teleconnections (Paluš, 2018). Bridge nodes spread a process to the entire spatial region globally whereas the effect of a local center is confined to a region (community) (Lawyer, 2015, refer Fig.2).

In temperature base climate networks it is the energy that is transported, and with this, some kind of information about the atmospheric state in a region (Hlinka et al., 2017). For rainfall networks in general, the links reflect the major propagation path ways of moisture, for extreme precipitation it is even more specific and reflects certain weather conditions, e.g. a specific "Großwetterlage" in central Europe. Ozturk et al., 2018 proposed a complex network based approach to estimate the tendency of extreme rainfall movement over Japan during typhoons. They iteratively approximated likely tracks of the extreme precipitation for each grid cell, many of which present redundant information, and hence the computation is time inefficient (several days). We suggest that by applying the same method only on global bridges and local centers, we can reduce

the redundancy in such large climate networks; and deduce the likely track of extreme events because individual grid points do not represent distinct climatological processes.

Kriging: I noticed that also the other reviewers asked for some improvements regarding the relative kriging errors. In addition to what they wrote, I was not sure whether the variogram is recomputed when stations are removed. If this is done, then variogram fitting is a science in itself, whether done manually or automatically, and this can lead to changes in the kriging error, making small changes more a result of random changes. The kriging error should normally not decrease when you remove stations, so the reduction in table 4 for the mean is most likely because the variogram has been fitted differently. When kriging error is used to estimate network modifications, the variogram is therefore usually kept constant, to avoid having to also analyze the variogram fitting. The larger changes are still significant.

We thank the reviewer for highlighting this important piece of information which is essential for the replicability of the work. However, we confirm that the variogram has been kept constant during the network modification. We will better explain the Kriging application in the revised version.

Some smaller issues:

P2L25 the sentence is somewhat contradictory to the previous one, try to rephrase.

Yes, the statements were contradictory, which will be modified in the revised version.

P13 - Fig 6 Remove 10% from the x-label

Will be changed in the revision.

P16 Eq. A3 explain why the numbers are 1 and ½ in the equation.

This definition of $J_{xy}$ prevents counting a synchronized event twice. When two synchronized events match exactly ($t_l^x = t_m^y$), we use a factor $1/2$ since it double counts in $C(x|y)$ and $C(y|x)$. We will add this explanation in the revision.

$$
J_{xy} = \begin{cases} 1 & if \quad 0 < t_l^x - t_m^y < \tau_{lm}^{xy} \\ \dfrac{1}{2} & if \quad t_l^x = t_m^y \\ 0 & else, \end{cases} \tag{A3}
$$

**References**

Hlinka, J., Jajcay, N., Hartman, D. and Paluš, M.: Smooth information flow in temperature climate network reflects mass transport, Chaos: An Interdisciplinary Journal of Nonlinear Science, 27(3), 035811, doi:10.1063/1.4978028, 2017.

Jensen, P., Morini, M., Karsai, M., Venturini, T., Vespignani, A., Jacomy, M., Cointet, J.-P., Mercklé, P. and Fleury, E.: Detecting global bridges in networks, Journal of Complex Networks, 4(3), 319–329, doi:10.1093/comnet/cnv022, 2016.

Lawyer, G.: Understanding the influence of all nodes in a network, Scientific Reports, 5(1), doi:10.1038/srep08665, 2015.

Ozturk, U., Marwan, N., Korup, O., Saito, H., Agarwal, A., Grossman, M. J., Zaiki, M. and Kurths, J.: Complex networks for tracking extreme rainfall during typhoons, Chaos: An Interdisciplinary Journal of Nonlinear Science, 28(7), 075301, doi:10.1063/1.5004480, 2018.

Paluš, M.: Linked by Dynamics: Wavelet-Based Mutual Information Rate as a Connectivity Measure and Scale-Specific Networks, in Advances in Nonlinear Geosciences, edited by A. A. Tsonis, pp. 427–463, Springer International Publishing, Cham., 2018.

---

## Author Response (AR1)

**Editor's comments**

Two referees have evaluated the paper "**hess-2018-113: Optimal Design of Hydrometric Station Networks Based on Complex Network Analysis".** Both referees mentioned that the paper presents an interesting and attractive idea, namely the transposition of complex network analysis methods to evaluate and support the optimal design of hydrometric networks. The authors proposed a ranking measure which helps apparently identifying the less influential stations. However both referees had made very important and fundamental critics, mainly doubts concerning the adequacy of the proposed method and the soundness of the conclusions.

We thank the editor for a constructive summary of our manuscript and also for his critical and supportive suggestions. Your feedback is vitally important to increase the readability of the work.

We have responded (in black) to each reviewer comment (in red).

Both referees had asked for additional explanations, justifications and analysis concerning:

The advantages of the method.

The advantages of the proposed method can be summarized under the following points and have been added to the revised manuscript:

- It has wide applications in all other relevant fields where identifying key nodes (influential nodes) in the network is vital for efficient and effective functioning of the system. Several methods have been proposed to evaluate the importance of nodes (Section 3.2). Our case studies show that the proposed measure WDB has an even higher discrimination power compared to existing node ranking measures and that it effectively ranks the nodes in the network. Additionally, WDB is sensitive to the different roles of nodes, such as global connecting nodes, hybrid nodes, and local centers, and provides a more informative ranking than the existing node ranking measures (P3/L8-13).
- Furthermore, our study emphasizes that the proposed approach for designing optimal observation networks, based on the complex network paradigm, is potentially more promising than existing ones (P18/L13-15).

The hypotheses used: e.g. the subjective cutoff value for the Qs to define the network, the stations to be removed first, the ranking method depending on the final objective.

In the revised version we have rectified all the above-mentioned issues (section 4.2, see response to below-mentioned reviewer-2 comments).

Improvements regarding the relative kriging errors.

We have carefully considered these critiques and responded correspondingly in the reviewer comments (Section 4.4).

The need for more tests for the validation;

As suggested, we have provided another test to check the interpolation error based on the leave-one-out sampling method providing more realistic estimates of real interpolation error (P16-17).

**Anonymous Referee #1**

I am not convinced that Fig.2 and Table 2 really help to understand what the result of the implementation of the proposed ranking method gives if applied to a real observation network. The example in Fig.2 is extremely simple and can hardly be extrapolated to a raingauge network including hundreds of gauges. It is unclear what node 5 in figure 2, which illustrates the added value of the proposed ranking method, really stands for.

We agree with the reviewer that the discussion of this simple example is not sufficient for explaining large observation networks, such as raingauge networks with hundreds of stations. However, our intention was to motivate the development and to illustrate the idea behind the new node ranking measure using a simple example. The differences to standard measures, such as degree and betweenness centrality, becomes more evident when looking at a small number of network nodes. Independently from this example, the proposition that our new measure is useful for large networks of raingauges is discussed and evaluated using the decline rate of network efficiency and the kriging error (section 4.3, 4.4).

 In the revision, we have made clear (P8/L5) that this simple example is used for easy understanding and motivation, and that we do not derive that our new measure works for the large raingauge network as well.

The role of node 5 in Fig. 3 (Fig.2. In earlier version) can be understood with the followings simple theoretical example (P8-9/L22).

In climate networks, local centers correspond to nodes which are important for local climate phenomena, while bridges correspond to nodes which connect different climatic subsystems, such as Indian monsoon and El-Nino, leading to teleconnections (Paluš, 2018). Bridge nodes spread a process to the entire network with more force than a local center in a community, even when the local center has more spreading power, i.e. more connections, than the bridging node (Lawyer, 2015). By deducing the global bridge node in a spatial hydrometeorological network, most of the process flow information can be captured.

What about raingauges that are poorly correlated with other gauges that could rather appear as dead-ends or even isolated nodes in the build network?

We interpret this question as a modified question of the reviewer in his/her first review: *"If I understand correctly stations with strong similarities with other stations will have a large number of connections and hence a high WDB value and conversely".* As we mentioned in our previous response letter the ranking measure that we propose (WDB) is not simply related to the number of links. Hence, the ranking of the nodes will depend on their information contribution to the network, see for example, on page 14, line 15: *"… it awards stations which provide unique information which cannot be generated from other stations in the network …"* However, we agree with the reviewer that such a situation needs to be considered carefully. We have added the following discussion on such situations in the revision (P10-11/L19).

There is one situation where our method would require additional care: Let's imagine a node that is unrelated to other nodes (no links). Physically, one might imagine this scenario in a meteorological sub-region characterized by fine-scale convective thunderstorms with sparse rain gauge coverage. Hence, precipitation event synchronization across rain gauges in that sub-region would be poor. In that case, indeed, this station would not be the part of the constructed network, and would not be ranked. This station should be treated carefully as it provides unique information.

Some extracted maps from figure 4 showing on a limited size area, the topography along with the location and resulting ranks of the raingauges and maybe also the location of the 10% higher ranked removed gauges could improve a lot the presentation of the method.

We thank the reviewer for her/his excellent suggestion. In the revised version we have incorporated a new figure showing 10% high ranking stations, 10% lowest ranking stations along with topography (Fig.9).

[Figure]

Figure 9: Location of 10% highest ranking rain gauges (red) and 10% lowest ranking stations (black) showing on Germany along with topography of the area.

Moreover, I confirm that additional validation is needed and this is acknowledged by the authors. At least the resulting variance of the rainfall fields corresponding to the various tested networks in table 4 should be provided. If significantly modified by the gauge selection method – probably moderately for the random selection method – it could have a major impact on the figures in table 4. This should be considered in the interpretations. But evaluations based on a leave-one-out approach should also be conducted.

As recommended by the reviewer in her/his review letter we have computed a new validation scheme based on leave-one-out approach and resulting variance of rainfall fields of corresponding various tested networks (Appendix A1) in the revised version.

"....We further check the interpolation error based on the leave-one-out sampling method providing another estimates of the interpolation error. For the three cases we eliminate successively stations from the network and study the impact on the relative kriging error. The leave-one-out approach is applied before the network reduction, i.e. on the original network of 1229 stations. For example, while removing highly influential stations from the entire network of 1229 stations, in the very first step, rank 1 station is eliminated and the relative kriging error is calculated for five characteristics, i.e., mean, 90%-, 95%-, 99%-percentile, and number of wet days (Fig. 8(a)-(e)). In the second step, rank 1 and rank 2 stations are eliminated from the network. We repeat the procedure up to 10 stations since the calculation is computationally very expensive.

We further repeat the procedure for the other two cases of expendable stations and random stations. All the results are plotted in the Fig. 8 (a)-(e). We observe that eliminating high ranking stations (influential) induce positive and high ($\Re > 5$%) error in the network as compared to removing expendable stations and random stations from the network.

Removing influential nodes in our 10 trials shows a smoothly increasing pattern of the error whereas the pattern for removing randomly selected stations is rather volatile. Here, we see sudden changes in the relative kriging error which is expected as stations with very different ranks can be removed from step to step. These results support our finding that eliminating highly ranked stations will have major impacts on the network, whereas low-ranking stations could be removed or relocated without inducing significant errors....."

[Figure]

Figure 8: Relative kriging error as a function of the number of stations removed from the network. In the first step we remove the highest ranking station, followed by the two highest ranking stations and so on. We then calculate the relative kriging error in the mean (a), 90th percentile (b), 95th percentile (c), 99th percentile (d) and wet days (e). We apply the same procedure to expendable stations and randomly selected stations.

**Remarks:** The real leave-one-out approach is different and used mainly for validation of the models. However, here the above done analysis (leave-one-out) only mimic what has been already showed by removing a bunch of raingauge from the network. Hence authors strongly suggest not to include this analysis in the revised version. Otherwise it might affect the readability of the manuscript.

We further provide additional details on kriging which has been added to the appendix A in the revised version.

**Kriging variogram modelling**

The kriging modelling mandates a theoretical variogram function that is to be fitted with an experimental variogram of the observed data. The experimental variogram ($\gamma(h)$) is calculated from the observed data as a function of the distance of separation (h) and is given by (Adhikary et al., 2015)

$$\gamma(h) = \frac{1}{2N(h)} \sum_{i=1}^{N(h)} [(Y(i) - Y(j))^2] \tag{A1}$$

where $N(h)$ is the number of sample data points separated by a distance $h$; , $i$ and j represent sampling locations separated by a distance h; $Y(i)$ and $Y(j)$ indicate values of the observed variable $Y$, measured at the corresponding locations $i$ and $j$ respectively. The theoretical variogram function ($\gamma * (h)$) allows the analytical estimation of variogram values for any distance and provides the unique solution for weights required for kriging interpolation.

The variogram models are a function of three parameters, known as the range, the sill, and the nugget (Fig.A1 (a)). The range is typically the level of h at the correlation between point values is zero (i.e., there is no longer any spatial autocorrelation). The value of $\gamma$ at range is called the sill. The variance of the sample is used as an estimate of the sill. Nugget represents measurement error and/or microscale variation at spatial scales that are too fine to detect and is seen as a discontinuity at the origin of the variogram model. The ratio of the nugget to the sill is known as the nugget effect, and may be interpreted as the percentage of variation in the data that is not spatial. The difference between the sill and the nugget is known as the partial sill.

The value of all the parameters and resultant variogram for mean, 90th percentile, 95th percentile, 99th percentile and wet days has been reported in the Table A1 and Figure A1 (b-d) respectively. The variogram has been kept constant during network reductions.

[Figure]

[Figure]

**Figure A1: Typical variogram models (a) and fitted variogram models for mean (b), 90th percentile (c), 95th percentile (d), 99th percentile (e) and wet days (f).**

**Table A1: Parameters values for the fitted variogram.**

| Parameters | Mean | 90th percentile | 95th percentile | 99th percentile | Wet days |
|---|---|---|---|---|---|
| **Nugget** | 0.0058 | 0 | 0 | 0 | 0.905 |
| **Range** | 0.0782 | 0.0782 | 0.0782 | 0.0782 | 2.363 |
| **Partial sill** | 0.103 | 1.055 | 2.140 | 6.808 | 2.771 |

**Anonymous Referee #2**

This manuscript introduced the use of complex network analyses for designing optimal hydrometric networks. I find the concept interesting, but the authors somewhat fail to explain **what the advantage of this method is,** and to make me really understand **what the network analyses will mean in the case of hydrometeorological observations.**
*It is clear how a linear network can be defined, as in Figure 1, but I find it difficult to imagine the network that is built from the event synchronization.*

We are sorry that we have not been clear enough in explaining the importance and usefulness of networks for climate data analysis. Here, we briefly explain the meaning of network analysis and its advantages in hydrometeorological observations. The explanation has been added in the revised version P2/L27.

Recently, complex network theory has emerged as a very powerful tool for understanding large spatiotemporal complex systems such as climate. In which a node represent a rainfall station, grid station, streamflow station etc. interconnected with links in a non-trivial manner. The links are computed based on statistical associations between climate parameter time series at different points on Earth. Hence, network analysis of hydrometeorological system have yielded new insights in the investigation of climate dynamics (Tsonis and Swanson 2008).

For example, Malik et al., (2012) used complex networks and event synchronization to investigate the spatiotemporal dynamics of the Indian summer monsoon. The analysis provided valuable insights into the spatial organization, scales, and structure of the rainfall events during the Indian summer monsoon (June–September). Marwan and Kurths, (2015) showed that complex networks can be successfully deployed to study the transitions in past climate or to develop a prediction scheme for extreme rainfall events. Climate networks have provided also insights regarding further important questions in climate sciences, such as stability of multidecadal oscillations, regional impact of teleconnections, heat transport in the climate systems, etc. (Tsonis et al., 2004, Gozalchiani et al., 2005, Donges et al., 2009, Palus et al., 2011, Berezin et al., 2012 ,Guez et al., 2012, Deza et al., 2013, Boers et al., 2013, Tupikina et al., 2014 and Feng and Dijkstra.,2014). Not only in the area of climate science, several other studies have demonstrated that complex network theory greatly contributes to the spatiotemporal analysis of brain dynamics, friction networks, traffic flow, turbulent heated jets and multiphase flow system, etc. Furthermore, Steinhaeuser et al., (2012) have shown that complex networks have the ability to extract the information content in multiple variables relevant to a variable of interest, thus leading to better understanding of the underlying process dynamics.

To explain the network construction using ES, we have added a new schematic figure in the revised version (Fig.2).

We have inserted the following schematic figure with modifications in a revised version to better explain the network construction using event synchronization (Section 2.2). All the equations and symbols have been explained in the main text of manuscript.

[Figure]

==Figure 2: Schematic of network construction using event synchronization (ES). All the equations and symbols have been explained in the main text.==

**Major comments**

*Additional synthetic case study for expandable stations: Maybe the authors could show a small example where only a few (imaginary) stations are analyzed with the network methodology. Then it can be shown how and why some stations are redundant and can be removed. The real case example from Germany is interesting, but with such a high number of stations, it is challenging to understand what actually happens.*

The specific application to use the WDB measure for ranking raingauges in Germany may indeed be difficult to understand. However, for easy understanding and visualization we have already included small examples (Fig.3 & 4) and that an additional small example would not bring additional benefit.

*Threshold cutoff justification: I am not convinced by the use of a somewhat subjective cutoff value for the Qs to define the network, without at least a much deeper discussion around the effect. This will to a large degree ignore the level of similarity, it is just a yes/no transformation. Increasing or decreasing the threshold could drastically change the importance of the nodes in the network. Two stations with similarity just above the threshold will be treated the same way as two stations which are almost identical. On the other hand, two stations just under the threshold are treated completely different than the stations just above, even if their similarities are almost the same.*

We thank the reviewer for raising the concern with the subjectivity of threshold. In the revised version we have rectified this limitation by selecting threshold objectively (Section 4.1).

=="…Here, $\boldsymbol{\theta}_{i,j}^{Q}$ is chosen in such a way to capture only highly synchronized stations. Two stations are significantly synchronized, if the synchronization value exceeds the 95th percentile of the synchronization obtained by two synthetic variables that have the same number of events positioned randomly in the time series. We calculate synchronization for 100 pairs of random time series from which we derive the 95th percentile of synchronization.== We use 95% as our threshold, since it satisfies both necessary conditions: a high synchronization and a sufficient number of extreme events for comparison (Agarwal et al., 2017; Rheinwalt et al., 2016; Stolbova et al., 2014).

Threshold cutoff (two stations just under the threshold are treated completely different than the stations just above, even if their similarities are almost the same) is a major common issue in all the methods for example, peak over threshold etc. Also, for instance, two person earning almost same amount might end up in paying different amount of taxes depending lying just above or below the prescribed threshold.

Global bridge node: The authors do several times mention the importance of global bridge nodes, and the possibilities these give in analyses of complex networks. For example: *"For instance, in climate networks an early warning signal could be generated by capturing the flow of information at such points."* This might be explained better in some of the references, but it should anyway be better explained what a local center and global bridge node really means in the climate network, and what kind of information we could particularly capture from this node.

In climate networks, local centers correspond to nodes which are important for local climate phenomena, while bridges correspond to nodes which connect different subsystems of climate (Jensen et al., 2016), such as the Asian monsoon and El Niño/Southern Oscillation, leading to teleconnections (Paluš, 2018). Bridge nodes spread a process to the entire spatial region globally whereas the effect of a local center is confined to a region (community) (Lawyer, 2015, refer Fig.2).

In temperature based climate networks it is the energy that is transported, and with this, some kind of information about the atmospheric state in a region (Hlinka et al., 2017). For rainfall networks in general, the links reflect the major propagation path ways of moisture, for extreme precipitation it is even more specific and reflects certain weather conditions, e.g. a specific "Großwetterlage" in central Europe. Ozturk et al., 2018 proposed a complex network based approach to estimate the tendency of extreme rainfall movement over Japan during typhoons. They iteratively approximated likely tracks of the extreme precipitation for each grid cell, many of which present redundant information, and hence the computation is time inefficient (several days). We suggest that by applying their method only on global bridges and local centers, we can deduce the likely track of extreme efficiently.

The same information has been added in the revised version in the P8-P9.

Kriging: I noticed that also the other reviewers asked for some improvements regarding the relative kriging errors. In addition to what they wrote, I was not sure whether the variogram is recomputed when stations are removed. If this is done, then variogram fitting is a science in itself, whether done manually or automatically, and this can lead to changes in the kriging error, making small changes more a result of random changes. The kriging error should normally not decrease when you remove stations, so the reduction in table 4 for the mean is most likely because the variogram has been fitted differently. When kriging error is used to estimate network modifications, the variogram is therefore usually kept constant, to avoid having to also analyze the variogram fitting. The larger changes are still significant.

We thank the reviewer for highlighting this important piece of information which is essential for the replicability of the work. We confirm that the variogram has been kept constant during the network modification and added this information to the manuscript (P16/L8).

Some smaller issues:

P2L25 the sentence is somewhat contradictory to the previous one, try to rephrase.

Yes, the statements were contradictory. We have modified them in the revised version.

P13 - Fig 6 Remove 10% from the x-label

Changed.

P16 Eq. A3 explain why the numbers are 1 and ½ in the equation.

This definition of $J_{xy}$ prevents counting a synchronized event twice. When two synchronized events match exactly ($t_l^x = t_m^y$), we use a factor $1/2$ since it double counts in $C(x|y)$ and $C(y|x)$. We have added this explanation in the revision (Section 2.2).

$$J_{xy} = \begin{cases} 1 & if \quad 0 < t_l^x - t_m^y < \tau_{lm}^{xy} \\ \frac{1}{2} & if \quad t_l^x = t_m^y \\ 0 & else, \end{cases} \tag{4}$$

[revised manuscript text omitted]

---

## Author Response (AR2)

**Editor's comments**

This is the second version of the manuscript describing how to use network analyses for optimizing the design of station networks. This is a novel approach for the ranking of stations of a measuring network that may be used for instance to improve a network structure or reduce the number of stations. Two reviewers have evaluated this revised version. Reviewer #1 mentioned that the presentation and validation of the proposed approach presents major defaults and limitations that have not been solved in this revised version, and asked to reject the manuscript. Reviewer #2 mentioned that the manuscript has been improved, but still asked for major revisions. The method and results deserve an in-depth critical analysis before they can be published. *I therefore would like to give a third and last chance for the authors to make substantial corrections in order to give responses to all points raised by both reviewers.*

We apologize that we haven't been able to clarify all issues in our previous revision. We also thank the editor for his critical and supportive suggestions. Your feedback is vitally important to increase the readability of the work.

We have responded (in black) to each reviewer comment (in red).

More detailed analysis and discussion on the ranking method and criterions, such as the characteristics of the stations with the resulting highest ranks (see Reviewer #1).

A detailed analysis on the individual stations resulting from ranking has been added to the revised manuscript (section 5, Fig.8).

Detailed analysis of the validation procedure (see Reviewer #1).

A detailed analysis of the two validation procedures (a) decline rate of network efficiency and (b) relative kriging error was presented initially. On being asked by the reviewer, we have further added a detailed discussion on validation using the leave-one-out approach in the first response letter (submitted on 21 Dec 2018, validated on 02 Jan 2019). We also remarked in the response letter that *"the real leave-one-out approach is different and used mainly for validation of the models. However, here the above-done analysis (leave-one-out) only mimic what has been already shown by removing a bunch of raingauge from the network. Hence authors strongly suggest not to include this analysis in the revised version. Otherwise, it might affect the readability of the manuscript"* on which we did, unfortunately, not get any response from the reviewer.

Further, a new detailed analysis regarding the theoretical kriging error has been added in the Appendix B.

Application of complex network analyses to a precipitation network (see Reviewer #2): further analysis on threshold effects, the methodology for network optimization and the added-value in comparison to other methods, uncertainty analysis, etc.

We are grateful to the reviewer for the detailed comments and suggestions, which have been adopted in the revised manuscript. Changes to the manuscript include detailed discussion on the thresholding (section 4.2), rephrasing text to be more easily comprehensible of methodology (section 2.3), and adding text on the advantages of the proposed methodology (section 1, p3/L3-25), a statement regarding data uncertainty (section 4.1). More specific changes are outlined below.

**Anonymous Referee #1**

The manuscript presents a novel approach for the ranking of stations of a measuring network that may be used for instance to improve a network structure or reduce the number of stations with the minimum loss of information.

We thank the reviewer for acknowledging the potential of the method in hydrology and his/her critical and supportive suggestions.

The presentation and validation of the proposed approach present nevertheless some major defaults and limitations that have not been solved in this revised version. First, the ranking method and criterions are defined without explicitly considering the objectives of this ranking procedure: identifying the stations providing the largest or the least additional information to the network.

We agree with the reviewer that the objective of the node ranking measure was not highlighted explicitly in the manuscript though it was mentioned several times implicitly with the help of the synthetic networks and interpreted results, see for example on page 18, line 17: *"… it awards stations which provide unique information which cannot be generated from other stations in the network …"*. In the revised version we have highlighted the objective of the node ranking measure explicitly in the abstract (P1—20-23) and in the introduction (P3-4/L33).

*"We propose a new node ranking measure, the weighted degree-betweenness (WDB), to identify the stations providing the largest additional information to the network. The highest ranks of the WDB-ordered raingauges correspond to the most influential stations in the network."*

As far as I could understand from the manuscript, the ranking is based to some kind of correlation measure – event synchronization. A priori, the stations that appear to be "synchronized" with the largest number of other stations will have the highest ranks. If it is so, the results will contradict the objectives: identify stations providing additional information. But, the proposed ranking method is more complex.

It seems that the reviewer has misunderstood the proposed WDB measure. The similarity measure *event synchronization* is used here to construct the rainfall network, which is the base for various network measures used to rank the nodes in the network. Hence, event synchronization is not used to derive the station ranking. To rule out the possibility of misunderstanding, we have added the following sentence to the manuscript (P14/L13-14):

This question repeats a previous concern of the reviewer in his/her first and second reviews: *"If I understand correctly stations with strong similarities with other stations will have a large number of connections and hence a high WDB value and conversely".* As we already stated in our previous response, the proposed ranking measure (WDB) is not simply related to the number of links. The ranking of the nodes will depend on their information contribution to the network as expressed by the betweenness centrality, see for example, on page 18, line 17: *"… it awards stations which provide unique information which cannot be generated from other stations in the network …"* The difference between the number of links and our new measure WBD is clearly presented through the artificial network example in Fig. 3. For instance, node 5 has the highest WDB score but the lowest decree score. This means that the most important node (according to our measure) has the lowest number of links in the network. In our revised manuscript, being asked by reviewer 1, we have highlighted a scenario (Fig. 3) which suggests that ranking is independent of the number of links.

The authors should at least (!!) try to analyze the characteristics of the stations with the resulting highest ranks. Figure 8 which has been added is one first, but not sufficient step into this direction.

In the revised version, we have analyzed rainfall statistical characteristics and network measures to highlight the characteristics of 10 % (~122) highest ranking stations (P18-19/L16-25, L1-10).

**Characteristics of high and low ranking stations**

*We further analyse the characteristics of the stations with the highest ranks. We plot the network (Figure 8a) corresponding to the 10% (~122) high ranking stations, i.e. all the links originating only from these 122 stations. The size and color of each diamond shaped raingauge mark their degree and betweenness. All other stations are plotted in the background without highlighting their degree and betweenness. This sub-network is still difficult to interpret, hence we further plot the connections corresponding to two high ranking stations (Figure 8b) and two low ranking stations (Figure 8c). Although the degree of these four stations is roughly the same, there is a striking difference in the connections between low and high ranking stations. The connections of low ranking stations are regionally confined, and they rather reflect the similarity in rainfall variability within (homogenous) regions. The plot of high ranking stations in Figure 8a highlights that high rank stations are not limited to high degree or betweenness stations. The latter measures represent the homogeneity (high degree = many similar nodes of similar dynamics) and the path in the network, respectively, whereas WDB represents the connectiveness. This could reflect the pathways of moisture transport or extreme rainfall propagation, or (in case of betweenness) the separation of similarly behaving regions (Molkenthin et al., 2015; Tupikina et al.,*

*2016). To test whether the long-range connections of the selected nodes in Figure 8b are a typical feature of high ranking stations, we compute the geographical distance between all the connected raingauges and plot its median (Figure 8d) and 95th percentile (Figure 8e) against the node ranking. There is a clear correlation between rank and distance: High ranking stations tend to show longer connections, implicitly affirming that the WDB measure has the potential to capture highly influential nodes in the network.*

[Figure]

*Figure 8: Connections and location of 10% (~122) highest ranking rain gauges (a). The size and color of the diamond marker indicate the degree and betweenness of the raingauge. Connections corresponding to two high ranking stations (b, station ID: 21320, 16149) and two low ranking stations (c, station ID: 26132, 20356). Median (d) and 95th percentile (e) geographical distance plotted against node ranking.*

Second, the validation procedure, based on estimated kriging standard error variances (or standard deviation) is not satisfactory (table 1). By the way, the definition of this kriging standard error is missing in the manuscript.

In the present study, the proposed ranking algorithm has been used on various synthetic and one real-world rainfall data. The results or specifically the ranking of raingauges has been checked with two different criteria

(i) decline rate of network efficiency, and (ii) relative kriging error in different hydrological parameters (mean, percentile and wet days). Both the criteria used for validation are in congruence with the interpreted results.

In the first revision, the reviewer asked to provide a more detailed analysis using the leave-one-out approach, though the reviewer agreed that leave-one-out is computationally expensive but necessary. In the first response letter we have performed a leave-one-out approach on the selected variables (mean rainfall, percentile rainfall and wet days) and described the results. We noticed that all the obtained results with the leave-one-out approach are in coherence with the previously reported results. However, the leave-one-out approach does not add new information and, hence, we suggest not to present the leave-one-out approach in the main text but in the appendix.

The detailed explanation of kriging and kriging standard error with the definition and mathematical formulas is already given in the first revised version of the manuscript (section 3.3.2, more specifically on page 13, line 10).

Since the variogram do not vary with the network and location, these theoretical kriging errors do only depend on the network structure (density and distribution of inter-distances between stations) and most probably also on the variance of the measured field for the considered variable. The authors should verify what is precisely computed in the kriging software they use.

Removing 10% of the stations from the network, reduces the network density, and should result in an increase of the "kriging error" with a magnitude provided by the random removal tests. An **increase** of the variance, as observed with the removal of the 10% lower ranked stations, is hardly possible except if the variance of the observed field is decreased. I suspect that the 10% lower ranked stations may correspond to stations with atypical measurements contributing significantly to the variance of the observed field: i.e. stations providing important additional information. If this is confirmed, the proposed ranking methods lead to a result that is exactly the opposite of its objectives.

We thank the reviewer for highlighting the need for discussion on the theoretical kriging error dependence. However, we had difficulty in understanding the reviewer's concern. In the above highlighted sentence "An increase (!) of the variance…." Does reviewer mean decrease?

However, based on our understanding we have tried to address the concern of the reviewer.

In the revised version we have added new discussion in section 3.3.2 and Appendix B. First, we have shown that the generated kriging model is spatially stationary and second we confirm that on removing stations either high ranking, low ranking or randomly selected the variance of the measured field will increase (not decrease!!). Also, with the help of a new added Fig.8 we have tried to explain why relative kriging error is insignificant on removing low ranking stations.

*Goovaerts (1997, p. 179) states that the theoretical kriging error is dependent on the covariance model and data configuration whereas it is independent of data values. In a given scenario of constant variogram during network modifications (as mentioned on P17/L7), theoretical kriging error only depends on data configuration (density and distribution of inter-distances between stations). To rule the possibility that these theoretical kriging errors also influenced by data values or spatial variance we double check the spatial stationarity of the measured field of the considered variable. Spatial stationarity means that local variation doesn't change in different areas of the map. For example, 2 data points 5 meters apart in different locations should have similar differences in your measured value. Kriging is not optimal for spatial abrupt changes and breaks lines. In literature, two methods have been proposed to check data's spatial stationarity with a Voronoi map symbolizing by entropy (variation between neighbours) or standard deviation and look for randomness in the measured field.*

*The first check has been performed in the ArcGIS (Geostatistical Analysis →Explore Data →Voronoi Map) on all the considered variables (mean, wet days, 90th-, 95th-, 99th percentile). The corresponding results for entropy Voronoi maps show the data set is looking adequately stationary (Appendix B).  However to quantify it, the second check has been performed in the Matlab using run test for randomness with the null hypothesis that the values in the data vector come in random order, against the alternative that they do not. The run test for randomness on all the considered variables rejects the null hypothesis with a p-value less than .0001. Hence, both methods confirm that kriging model used is spatially stationary, one of the mandatory condition to perform kriging.*

[Figure]

**Figure B1: Representation of mean (a), 90th percentile (b), 95th percentile (c), 99th percentile (d) and wet days (d) spatial stationarity with a Voronoi's map symbolising by entropy (variation between neighbours) using ArcGIS.**

Further, we would like to bring to attention that the main motivation was to see, how removing the low ranking and high ranking stations impact the kriging error variance. Hence in the used kriging software, we have *computed the square root of the kriging variance, also named as kriging standard error (KSE) (P12/L14) corresponding to the statistical rainfall measures (mean, wet days, and percentile rainfall) for the original network and reduced network.* ==We confirm that on removing stations either high ranking, low ranking or randomly selected the variance of the measured field will increase (not decrease!!).== *Intuitively, "the kriging variance is expected to be greater at a location surrounded by data that are very different from one another (Fig. 8b) than at a location surrounded by similarly valued (Fig. 8c) data" (Goovaerts, 1997;* Heuvelink and Pebesma, 2002*). And hence, we notice high kriging error in case of influential stations comparative to random and low ranking stations.* We further stress out that relative kriging error in case of removing low ranking stations are insignificant rather than highlighting that it has decreased, as mentioned earlier (corresponding text has been changed in the discussion).

This is the reason why I asked the authors to provide the variance of the various tested fields in table 1 and to conduct leave-one-out tests to compute real interpolation errors for the various tested networks. None of these important results have been provided in the revised version or in the responses of the authors.

We are surprised by this statement. In the submitted response letter from 21 Dec 2018 validated on 02 Jan 2019, we have considered both suggestions and presented the corresponding results in the highlighted text.

We also remarked in the previous response letter that *"the real leave-one-out approach is different and used mainly for validation of the models. However, here the above-done analysis (leave-one-out) only mimics what has been already shown by removing a bunch of raingauge from the network. We would again like to highlight that the kriging and network degree would only measure the homogenous regions, but WDB measures the connectivity (moisture pathways and separations between different regions) that are not identifiable with the kriging method. Hence authors strongly suggest not to include this analysis in the revised version. Otherwise, it might affect the readability of the manuscript"* on which we did not get any response from the reviewer.

To my opinion, the presented method and results may be interesting, but raise some major questions that have not been answered by the authors in the revised version of their manuscript. The method and results deserve an in-depth critical analysis before they can be published. I therefore suggest rejecting the manuscript in its present form.

In our opinion, the doubts raised are on the one hand a consequence of a misunderstanding by the reviewer. In the revised version the confusion between the two measures degree (number of connections) and WDB has been rectified by being even more explicit in the description of the method (Fig. 3). On the other hand, we feel that the reviewer is mostly concerned with the kriging method. In the submitted response letter from 21 Dec 2018 we have considered all the suggestions leave-one-out approach, variogram modelling (Appendix A),

influence of variance on theoretical kriging error (Appendix B) and detailed representation of the station resulting from high rank (Fig.8) and corresponding results are presented in discussion sections.

We also would like to highlight to the reviewer that the kriging has been used as an additional validation measure for the proposed ranking measure. It is not the core aspect of the work and, hence, being too focused (critical) on the kriging method would deviate from the major objective of the study. It is worth mentioning here that the kriging and network degree would only measure the homogenous regions, but WDB measures the connectivity (moisture pathways + separation of regions) that are not identifiable with the kriging method.

**Anonymous Referee #2**

This is the second version of the manuscript describing how to use network analyses for optimizing the design of station networks. The manuscript has been improved, and although I am far from convinced by the application of the methodology in hydrology, I think the idea and the application still has a novelty that makes it worth publishing after another revision.

We thank the reviewer for acknowledging the efforts of the authors and constructive summary of our manuscript and also for his critical and supportive suggestions. Your feedback is highly appreciated.

This manuscript has two parts:

- The new node ranking measure, which can be applied to any network analyses
- Application of complex network analyses (and the new node ranking measure) to a precipitation network.

For the first part, it seems the authors have introduced a new and improved measure, and although not a network expert myself, I think they have documented well the usefulness. For the second part, I am still not convinced by the usefulness of the methodology for meteorological networks. However, I think the idea of using it here is interesting enough to merit publication anyway. I have a few suggestions below for analyses and figures that could convince me more, but some of them might fit better in a follow-up paper.

The authors again appreciate the detailed comments and suggestions, which have been adopted in the revised manuscript.

Changes to the manuscript include the rewriting of sentences to make easily comprehensible, a more detailed discussion on thresholding issue, network measures. More specific changes are outlined below.

Figs 3 and 4 appears to show a spatial network. However, the networks analyzed here are made from correlations, therefore it would be interesting to see this transformation, and how spatial patterns of station proximity are kept or distorted.

The examples presented in Fig. 3 and 4 are synthetic networks which can be used to demonstrate any specific interactions of entities such as in a citation network, coauthor network or friendship network. They do not represent spatial patterns. In the present study they are used to explain the interaction of raingauges, based on observed data, which is indeed a spatial network. Physically the link between two raingauge nodes corresponds to the shared/common climate forcing which derives them. It is important to note that for the network construction the geographical proximity is not taken into account.

However, there are certain scenarios where the geographical proximity could be important and need to be properly accounted for. For instance, a study dealing with grid data of the entire globe where the distance between grid points at the poles is significantly lower than in the equatorial region. To account for the bias caused by the geographical proximity, so-called node splitting network measures have been proposed (Heitzig et al., 2012). However, the considered region and stations in our study is far from such scenario and hence this issue is not highlighted in the manuscript.

The authors mention the threshold effect in the answers, but I don't think it is enough just to mention that threshold effects also exist in other fields. I think this is an issue that should be added to the discussion of the manuscript.

We thank the reviewer for highlighting the need for discussion on thresholding issue. In the revised version we have added and highlighted the thresholding issue in section 4.2

*In literature, Two criteria have been proposed to generate an adjacency matrix from a similarity matrix, such as fixed amount of link density (Agarwal et al., 2018a; Donges et al., 2009a; Stolbova et al., 2014) or global fixed thresholds (Jha et al., 2015; Sivakumar and Woldemeskel, 2014). However, both criteria are subjective and may lead to the presence of weak and non-significant links in the adjacency matrix or network. These non-significant links might obscure the topology of strong and significant connections, Hence, stringent threshold criteria are needed, such as multiple testing (Agarwal, 2019; Boers et al., 2019). Alternatively, networks should be characterized across a broad range of thresholds. Furthermore, all self-connections or negative connections (anti-correlation), if any, should be removed (Rubinov and Sporns, 2010).*

*To minimize these threshold effects, we choose the threshold $\theta_{i,j}^Q$ objectively by considering all links in the network that are significant. A link is significant (i.e. two stations are significantly synchronized) if the synchronization value exceeds the 95th percentile of the synchronization obtained by two synthetic variables that have the same number of events positioned randomly in the time series. We calculate synchronization for 100 pairs of random time series from which we derive the 95th percentile of synchronization. Using a 5% significance level, we assume that synchronization cannot be explained by chance, if the ES value between two stations is larger than the 95th percentile of the test distribution. Here, we select 5% significance level since it is the well accepted criteria for the network construction.*

The methodology is presented as an alternative method for network optimization. But there is little discussion about the alternatives, and what advantages this particular method could provide. An alternative method is to optimize based on the kriging error. My guess is that the proposed method might be better for observations which are not only connected by spatial proximity, or where the effect of spatial proximity differs in different part of the study domain, but this would have to be properly described/discussed/analyzed.

We agree that more details about the advantages of the complex network approach are needed. The details are now mentioned in the revised version on page 3, line 3.

*It is now highlighted that "We use complex networks since it is a powerful approach in extracting information from large high-dimensional hydrological datasets (Donges et al., 2009a; Cohen and Havlin 2010). This non-parametric method allows investigating the topology of local and non-local statistical interrelationships. An example for non-local connections in a climate network are global influence of El Nino Southern Oscillation (ENSO) on rainfall (Agarwal, 2019; Ferster et al., 2018) and Atlantic Meridional Overturning Circulation (AMOC) on air surface temperature (Caesar et al., 2018) via teleconnections and ocean circulation respectively. The method allows to represent the dataset in form of spatially embedded network and visualize the connections. Once the spatial network of stations is set up, one might use network measures (e.g. degree, betweenness centrality) to analyse a range of aspects, such as community structure unravelling dominant climate modes (Agarwal et al., 2018a; Fang et al., 2017; Halverson and Fleming, 2015; Tsonis et al., 2011), catchment classification indicating hydrologic similarity (Fang et al., 2017), short and long-range spatial connections in rainfall (Agarwal et al., 2018a; Boers et al., 2014b; Jha et al., 2015; Stolbova et al., 2014) and spatio-temporal hydrologic patterns (Halverson and Fleming, 2015; Konapala and Mishra, 2017). Further, a recent study by Donges et al., (2015) pinpoints that complex network analysis can complement classical Eigen techniques, such as empirical orthogonal functions (EOFs) or coupled patterns (CP) maximum covariance analysis. They showed that EOFs, CPs and related methods rely on dimensionality reduction, whereas network techniques allow studying the full complexity of the statistical interdependences structure and are not limited to linear and spatial-proximity connections. Also, it has been shown that higher-order complex network measures (betweenness centrality, closeness centrality, participation coefficient) provide additional information on the higher-order structure of statistical interrelationships in climatological data (Donges et al., 2015). For example, the network degree represents similar information as the first eigenmode of an EOF analysis (Donges et al, 2015), in-out degree (incoming links and outgoing links in directed network) represents spatial-temporal propagation (Boers et al., 2014a), and betweenness centrality represents a spatial separation between flows or a handful of stations which are positioned in-between the large communities. These stations belong to large communities, but unlike most stations they tend to possess intercommunity connections ((Halverson and Fleming, 2015; Molkenthin et al., 2015; Tupikina et al., 2016).*

Measurement uncertainty seems not to be mentioned at all. This would most likely have an effect of the results. I can understand that this might be too complex to address properly in this manuscript, but at least the assumption about no measurement errors should be added to the text somewhere.

We have added this assumption in section 4.1 (P13/L26) which reads that "data processing and quality control were performed according to Österle et al. (2006), and in this study we assume that data is free from measurement errors".

I think Fig 2 is somewhat unclear. If I understand it right, the temporal distance between events in the two figures in the middle should reflect the first figure, but I don't see that this is the case.

We thank the reviewer for noticing a plotting mistake, which has been rectified with the new figure.

[Figure]

Figure 2: Schematic of network construction using event synchronization (ES). Equations and symbols are given in the main text.

In 2.3, it is somewhat unclear which of the measures that are part of the methodology, and which are the measures that are used for comparison. Maybe Fig 3 could be used for explaining some of the concepts here?

We have modified the section 2.3 clearly listing the traditional and contemporary network measures (P6-7/L11-L1-3). Traditional network measures which are the part of the methodology has been discussed in the Fig.3 following which Fig.4 compare the WDB with the contemporary network measure.

The wording seems to be based on the concept of network analyses. For example on P8, L17, it is referred to flow of information. What is really the flow of information in this case? Contrary to computer networks or social networks, the correlation of a variable between two observation locations is independent of the "flow" through a station between them. I think the authors should really consider the wording when they describe a network that is based on correlation rather than physical nodes with direct links.

The reviewer has highlighted an important point. Indeed, a network reconstructed by correlation measures, as in our case, often has no physical equivalence. A link represents a statistically similar behavior between

nodes, but not necessarily a real physical link. In some applications, such a reconstructed link can represent a real physical link, such in brain networks. In climate networks, the network topology is a useful tool to statistically analyze spatio-temporal patterns. Network measures represent certain aspects of the spatio-temporal field. For example, the network degree represents similar information as the first eigenmode of a EOF analysis (Donges et al, 2015), in-out degree (incoming links and outgoing links in directed network) represents spatial-temporal propagation (Boers et al., 2014) , and betweenness centrality represents a handful of stations which are positioned in-between the large communities and unlike most stations they tend to possess intercommunity connections (Halverson and Fleming, 2015; Molkenthin et al., 2015; Tupikina et al., 2016). The proposed measure WDB represents the importance of nodes for the information flow within the general context of complex networks. Within the specific application on rainfall networks, it describes a very specific property of the spatio-temporal patterns: the importance of a certain location for the exact description of the spatio-temporal patterns, such as boundaries between differently behaving regions. Again, WDB does not emphasize such locations within a homogenous region, but more the differences between homogenous regions as shown in Fig. 8. Therefore, it differs from other methods such as kriging.

Regarding the kriging error, it is only written that that this is estimated "across the entire study area". I would expect "across the study area" to mean a grid covering Germany, but the authors should add some more details about how many and how they selected the evaluation locations.

The word entire has been deleted (section 3.3.2, P12/L18).

Further details "how many and how we have selected evaluation locations" asked by the reviewer are indeed important, however, this section was dedicated to only introduce kriging briefly. All such information (highlighted below) is already existing in section 4.4.

The rain gauges are ranked in decreasing order according to their WDB values. Highly ranked rain gauges are interpreted as the most influential stations, and low ranked as expendable stations. Three cases are investigated: removing the 10% (total 122 stations) highest ranking stations, removing the 10% lowest ranking stations, and ten trials of removing 10% of the stations randomly.

P17L10-11 I had a quick look at the cited paper, and I don't see that they describe how M out of N stations can reduce the uncertainty.

We apologize for the writing style which confused the reviewer. We have changed the text as follows (P17/L22).

The study by Villarini et al., (2008) proposed a simple rule for the number of rain gauges required to estimate areal rainfall with a prescribed accuracy. In such scenarios, WDB measure could be applied to identify prescribed number of influential stations.

***Minor edits***

P3L14 "This interactionS is" - remove S or change This is.

Corrected (P3/L21).

P4L5 remove space after Bullmore and Sporns, 2012)

Corrected.

P7L4 hybrid nodes are poorly defined, maybe make a reference to Fig 3 here?

Done.

P9L4 I'm sure there is a fairly good English translation for Grosswetterlage.

Done.

Fig 3 caption Most of this is a description that belongs (and mostly already is) in the main text.

Authors agree with the reviewer and correspondingly caption has been shortened in the revised version.

Many places: I'd say that the "relative kriging error" should rather be referred to as something like the "relative increase of kriging error".

Done.

P17L9 "in the region,"

Corrected.

P17 Caption Fig 8 "showing on Germany" – should be rephrased

Done.

P19L5-6 This sentence is first of all unclear. Additionally, many variogram models have a range parameter that is not equal to a distance where the correlation is zero.

Corrected.

[revised manuscript text omitted]

---

## Author Response (AR3)

**Editor's comments**

This is the third version of the manuscript. The manuscript has been further improved, but there are still some issues as mentioned by the reviewer. I would like to give the authors a chance for a last iteration to improve it. The paper will be sent again for review. Please make significant changes and give detailed responses to each point raised by the reviewer.

We greatly appreciate the opportunity to resubmit the study and the referees' concerning improvement to this paper. We have revised the manuscript according to reviewer's comments and provide our responses point to point as followed.

*Major revisions include*

1. Inclusion of an example of the transition from observations of a very reduced measurement network to a complex network, and how the nodes are ranked, using proposed WDB measure, in this network [*Appendix A*].

2. Moderating tone of the paper to present the current state of the methodology and the analysis in the preliminary phase (Highlighting the limitations and future scope of the work at various places as highlighted in the manuscript and response letter)

3. Explanation of kriging evaluation

4. Avoiding terminology of a physical network and acknowledging the fact complex networks are in infancy state at least in hydrology and more interpretable measures are needed.

5. Figure modifications

We have responded (in black) to each reviewer's comment (in red).

**Anonymous Referee #1**

*General summary*

This is the third version of a manuscript describing how to use network analyses for optimizing the design of station networks. The manuscript has been further improved, but there are still a few open issues. I'm still not completely convinced by the methodology, but as also mentioned in my previous reviews, I like the novelty of the approach and the attempt to think out-of-the-box, and think the authors should get a last chance to convince us that it deserves publication. I have some suggestions below on how the authors can make this publishable, even though we are not convinced by the usefulness.

We acknowledge that we haven't been able to clarify all issues in our previous revisions. We thank the reviewer(s) for his critical and supportive suggestions. Your feedback is vitally important to increase the readability of the work.

I think the authors should present the current state of the methodology and the analysis as less mature than what they now try to conclude, as I think follow-up studies are necessary before the methodology can be considered a trustworthy optimization tool for measurement networks.

We very much appreciate the suggestion of the reviewer. In the revised version we have moderated the tone throughout the paper. For instance,

*P1/L24-25: Although, this is the first step and the real benefit of the method needs to be investigated in more detail.*

*P3/L13:* however, the application and interpretation of complex networks in hydrology are in the infancy state.

P3/L19-*23: We do acknowledge that this study is preliminary efforts to explore complex networks application in hydrology and many further studies are necessary before the methodology can be considered a trustworthy optimization tool for measurement networks.*

*P20/L15-19: However, acknowledging the preliminary work done in this study, WDB application needs to be investigated in detail and currently out of scope of the study domain. In addition, follow-up studies addressing threshold and spatial boundary issues of the network, physical interpretable measures and visualization are needed to prove the benefit of complex networks science in hydrometric network design.*

As mentioned before, I would like to see an example of the transition from observations of a very reduced measurement network to a complex network, and how the nodes will be ranked in this network. Ideally, this should show the locations of (maybe) 6-10 stations with a table indicating their cross-correlations, and the resulting network. If this adds too much to the paper, it should be added to the appendix. Figure 8 is interesting, but it does not answer the question I have asked previously. Such a figure would also be an answer to the first comments of Reviewer #1, who also seems to find it difficult to visualize how the methodology will identify redundant stations or stations that provide additional information. The analysis of 1229 stations is nice to see the overall effect, but it does not help us to understand the method.

We thank the reviewer for highlighting the need to show the much reduced spatially embedded network. In the revised version we have added the reduced network consisting of 11 rain gauges and corresponding locations and cross-correlations in Appendix A.

**A. Spatially embedded network construction**

Further, to illustrate the network construction from observations of a very reduced measurement network to a complex network, we select randomly 11 rain gauge stations spread across the Germany. The geographical locations of these stations (Table S1) are shown in Fig. S1 (a). We first compute the cross-correlation between each pair of two stations (Table S1) and then apply 90$^{th}$ percentile threshold. Links exist between pair of stations having correlation value greater than threshold (Fig. S1 (a)).

We further compute the WDB score for each station using Eq. 10 (Fig. S1 (b)). Stations 3 shows the highest WDB score in this particular network consisting of 11 stations which signifies that station 3 not only account the local and global characteristics of this particular stations but also the cumulative effect of the influence or contribution of the directly connected stations. For instance, it seems like two strong modular (homogeneous) regions (stations 1,2,3, 6 & 8 and 3,4,5,7,9,10 & 11) are present within the network bridged by station 3. This node particularly very important in a measurement network in the context of measuring process, process

identification or interpolation of measurements. For instance, at the particular location two different processes might be dominating (snow and rainfall). Other interpretation could be that the implicit assumption of the complex network is that station 3 is representative for a larger area than other stations. But again, it is challenging to quantify at this stage and indeed follow up studies are needed to prove the benefit of complex networks science in hydrometric network design.

Table S1: Cross-correlation values along with the geographical location of ten rain gauges selected for network illustrative purposes.

| Nodes | Lat. | Long. | 1 | 2 | 3 | 4 | 5 | 6 | 7 | 8 | 9 | 10 | 11 |
|---|---|---|---|---|---|---|---|---|---|---|---|---|---|
| 1 | 1.00 | 0.46 | 0.50 | 0.32 | 0.33 | 0.59 | 0.41 | 0.42 | 0.27 | 0.32 | 0.24 | 1.00 | 0.46 |
| 2 | 0.46 | 1.00 | 0.58 | 0.38 | 0.38 | 0.43 | 0.39 | 0.54 | 0.30 | 0.40 | 0.27 | 0.46 | 1.00 |
| 3 | 0.50 | 0.58 | 1.00 | 0.41 | 0.51 | 0.45 | 0.49 | 0.48 | 0.35 | 0.50 | 0.36 | 0.50 | 0.58 |
| 4 | 0.32 | 0.38 | 0.41 | 1.00 | 0.45 | 0.27 | 0.30 | 0.31 | 0.27 | 0.41 | 0.29 | 0.32 | 0.38 |
| 5 | 0.33 | 0.38 | 0.51 | 0.45 | 1.00 | 0.30 | 0.41 | 0.33 | 0.40 | 0.64 | 0.46 | 0.33 | 0.38 |
| 6 | 0.59 | 0.43 | 0.45 | 0.27 | 0.30 | 1.00 | 0.39 | 0.44 | 0.24 | 0.30 | 0.22 | 0.59 | 0.43 |
| 7 | 0.41 | 0.39 | 0.49 | 0.30 | 0.41 | 0.39 | 1.00 | 0.39 | 0.52 | 0.45 | 0.41 | 0.41 | 0.39 |
| 8 | 0.42 | 0.54 | 0.48 | 0.31 | 0.33 | 0.44 | 0.39 | 1.00 | 0.29 | 0.37 | 0.25 | 0.42 | 0.54 |
| 9 | 0.27 | 0.30 | 0.35 | 0.27 | 0.40 | 0.24 | 0.52 | 0.29 | 1.00 | 0.46 | 0.51 | 0.27 | 0.30 |
| 10 | 0.32 | 0.40 | 0.50 | 0.41 | 0.64 | 0.30 | 0.45 | 0.37 | 0.46 | 1.00 | 0.50 | 0.32 | 0.40 |
| 11 | 0.24 | 0.27 | 0.36 | 0.29 | 0.46 | 0.22 | 0.41 | 0.25 | 0.51 | 0.50 | 1.00 | 0.24 | 0.27 |

(a)

[Figure]

(b)

[Figure]

Figure S1: Sample rain gauge network constructed using cross-correlaton similairy measure and 90th percentile threshold only for illustrative purproses. Autocorrelation (digonal) has been ignored in the network costruction. Numbers 1 to 11 are node counts, and values in brackets represent the WDB values.

There is still no information about the evaluation locations for finding the kriging error. The author's response refers to section 4.4, but no, it is not there. And the fact that there is still a decrease in kriging error for case II indicates that something in the setup does not follow the standard procedures for estimating the effect of removing stations on kriging error, even if the authors have now removed the emphasize on this in the text.

We apologize to the reviewer for indicating to the wrong section regarding the kriging evaluation process in the previous response letter. For finding kriging error for all 3 cases, evaluation has been across the study area. This has been mentioned clearly in the revised version (section 4.4).

*"As the second approach to assess the suitability of WDB for identifying influential and expendable stations, we analyze the change in the kriging error when stations are removed from the network. We first estimate the kriging standard error across the study area for all 1229 stations termed as $KSE_{old}$. Then, we measure the increase or decrease in the kriging standard error across the study area when stations are removed terms as $K_{new}$. "*

In our previous version, insignificant negative values were observed for case II. We have thoroughly checked the model and have run it for 100 times. *"P16/L15: For each case and rainfall characteristics we run model 100times and the mean value of $\Re$ has been reported in Table 1."* We noticed that the outcomes are robust for the given scenario. However, in the future, to further advance the model weighted kriging method could be used.

I see that the authors still use phrases as "global bridge nodes are able to control the flow of information" (P20). Being a measurement network, a node does not control anything. Nodes that are identified as bridge nodes might be particularly important also in a measurement network, but the authors should manage to describe the particular usefulness of these nodes in the context of measuring, process identification or interpolation of measurements, not use the terminology of a physical network. The authors mentions "Information flow, or in our context, transferability of precipitation measurement across locations, would be restricted in their absence" on P9L15-17.

This is a start, but should probably be better emphasized, and not only on this location.

The authors again appreciate the detailed comments and suggestions, which have been adopted in the revised manuscript.

All terminologies related to physical networks are avoided up to much extent. Indeed, as the reviewer suggested complex network is in infancy state in hydrology and physical interoperation of network measures is rather crude. This also is in congruence with the reviewer's first suggestion *"authors should present the current state of the methodology and the analysis as less mature than what they now try to conclude."* Changes in the revised version are as follows:

*P9/L1-11: In general, high degree nodes represent most connected (highly correlated) nodes in a network. Rheinwalt et al., (2015) considered these highly correlated nodes of homogeneous precipitation community as local centers representing homogenous precipitation patterns for that particular community. Agarwal et al., (2018) defined local centres as the nodes having maximum intra-community links and minimum inter-community links based on the Z-P space approach. However, degree alone cannot distinguish the roles of nodes in the sample network as seen for nodes 5, 7, and 8, which have the same degree ($k_i=2$), though node 5 serves as a bridge node linking the two parts of the network. In a larger complex network, such bridge nodes have strategic relevance as most of the information can be accessed quickly just by capturing those nodes. For example, Kurths et al., (2019) quantified the spatial diversity of Indian rainfall teleconnections at different timescale by identifying linkages between climatic indices (e.g. El Niño/Southern Oscillation, Indian Ocean Dipole, North Atlantic Oscillation, Pacific Decadal Oscillation, and Atlantic Multidecadal Oscillation) and seven Indian rainfall stations (bridge nodes).*

*P9/L13-17: As mentioned, global bridges connect different parts of a network (e.g. teleconnection between Indian rainfall and ENSO) and measuring and interpretation of spatially large variations, process identification, interpolation of measurements and transferability of precipitation measurements across locations, would be restricted in the absence of high $B_i$ nodes.*

*The discussion and conclusions are mainly a summary of the results, with some very positive interpretations of the results and the usability for measurement networks. Whereas I understand why this is done, and I know that this is a rather typical end of a manuscript, I think the authors need to reflect a bit more about advantages and disadvantages of testing this methodology within measurement networks. I don't think the suggested figures above would completely convince me, so it would be much easier to accept a manuscript that also includes some doubts and constructive self-criticism. We are two reviewers who have presented the authors with many uncertainties and doubts. I think the authors would strengthen the manuscript by trying to further address some of these in the discussion, not necessarily giving a good answer to all, but accept that the methodology still has its uncertainties in this context. A considerable amount of work would still be necessary to convince me that this is really the way forward, but I'm happy if the conclusions just prepare the ground for these in forthcoming work.*

We thank the reviewer for suggesting to highlight the limitations and future scope of the work. We have added most of the details all over the manuscript and in conclusion as well.

*P20/L20-25: "However, acknowledging the preliminary work done in this study, WDB application needs to be investigated in detail and currently this is out of the scope of current study domain. In addition, follow-up studies addressing threshold and spatial boundary issues of the network, physical interpretable measures and visualization are needed to prove the benefit of complex networks science in hydrometric network design."*

*Seeing Figure 4 again, I wonder if the authors could comment on the following assumption that would be somewhat intuitive for a geoscientist, although it might not be correct:*

Node 7 is correlated with four other stations, and its value could most likely be replaced by the interpolated value without much loss of information. Stations 5, 10 and 11 are each correlated with one station only, and might have more unique information than station 7. Maybe the implicit assumption of the complex network is that station 7 will be representative for a larger area than station 10? But again, it is challenging to understand the choices and consequences of using this new methodology for measurement networks without a figure as the one mentioned under 1, further up.

We thank the reviewer for highlighting the need to understand the implicit assumption of the complex network which I am sure is rather difficult at this stage. We indeed need better understanding of network measures and visualization tools as highlighted in the conclusion of the revised version (P20/L19-24). Our ultimate argument is that that WDB quantifies the uniqueness (or redundancy) of the information which a station provides. We have interpreted this statement directly from the equation the way how WDB is calculated.

*P17/L8-12: The proposed WDB method based on degree and betweenness centrality not only account the local (captured by degree) and global (captured by betweenness centrality) characteristics of nodes but also the cumulative effect of the influence or contribution of the directly connected (localized) nodes.*

I find it hard to visually compare the values of the measures in Figs 3-4. I think it would be easier if the numbers were ordered in a column instead of a row. Alternatives would be more space between the commas and next number, or different color for each measure.

We have modified Fig.1, 2 and 3 as suggested.

[Figure]

Figure 1: Topology of two sample networks to explain network structures and measures. (a) Network N1 with four nodes and three links; (b) network N2 with four nodes and six links.

[Figure]

Figure 2: Synthetic network to explain the degree *(k)*, betweenness centrality ($B$) and weighted degree-betweenness ($WDB$) measures, with node number (1 to 8) followed by the degree, betweenness centrality value and $WDB$ value in brackets *[k, B, WDB].* Degree and betweenness are limited in distinguishing the role of different nodes in the network and centers from bridges, respectively.

[Figure]

Figure 3: Synthetic network used to compare the network measures betweenness centrality, bridgeness, and DIL with the proposed measure WDB. Numbers 1 to 11 are node counts, and values in brackets represent the network measure values in order of $[B, Bri, DIL, and \ WDB]$. Node 6 is a global bridge node that connects two sub-networks. Node 4 and 7 are hubs that are connected to most of the nodes in the sub-networks. Node 5, 10 and 11 are the dead-end nodes.

***Minor edits***

P1L14 though -> although?

Done

P1L26 rain GAUGE stations?

Done

P2L31 "We use complex network which" -> something seems to be missing

We have revised the sentence.

*"We propose complex networks as a suitable tool for this optimization problem. A complex network is defined as a collection of nodes, such as rain gauge stations, interconnected with links. "*

P3L5 Check grammar: An example … are …

Done

P3L8 comma before respectively

Done

P8L2-3 the term "except node 3" is ambiguous, it could also mean that Bri != 0 for Node 3. Rephrase?

We have revised the sentence.

*"The neighbourhood of node i ($N_G(i)$) consists of all direct neighbours of node i. For example, in the networks N1 and N2, all nodes (except node 3 in N1) have $B = 0$ hence $Bri = 0$. However, node 3 in the network N1 has all the nodes in direct neighbourhood hence, it also has $Bri = 0$."*

P19L4 high degree or betweenness stations -> doesn't read well, something missing?

We have revised the sentence.

*"Highest ranked stations are not governed by only local or global features but rather the quantitative combination of both (Figure 7a)."*

P19L6-7 two times "or"

Deleted second repetitive "or"

P19L14 Fig. 8 IS also ...?

Done

P19L15-19 It is not clear how the citation relates to what comes after. Something missing? Is this referring to two different areas in the same modelling region that, despite different local variance, will have the same modelled kriging error (function of the variogram model and configuration of neighboring observations), or is it about two modelling regions with different variogram and different error structure?

We have modified the sentence.

[revised manuscript text omitted]

---

## Author Response (AR4)

**Editor's comments**

This is definitely an improved version of the manuscript describing how it could be possible to use complex network analyses for designing a hydrometric station network. I think this can be published with a few minor corrections. Some copy-editing is necessary, particularly of the newly added text.

We greatly appreciate the opportunity to revise the manuscript and the very helpful comments of the referee. We have revised the manuscript according to the reviewer's comments and provide our responses point by point as follows.

We have responded (in black) to each reviewer's comment (in red).

**Anonymous Referee #1**

I'm happy to see the appendix A, but some modifications are needed.

1. There are no references to the appendix (neither A or B) in the text, please add.

   Several references have been added in both the appendix A and B.

2. The actual threshold used (0.5?) could be mentioned explicitly, e.g. on P21L11.

   Done.

   *Links exist between pairs of stations having a correlation value greater than 0.44 (Fig. A1 (a)).*

3. The text should be edited; there are many errors and imprecise phrases. (stationS 3, comma before which, "this node particularly very important"?, ...)

   We have edited the text in the revised version.

4. It seems coordinates in the table are missing, and the correlation matrix has been shifted.

   Corrected.

5. I think "Autocorrelation ..." in the caption can be removed, seems quite obvious.

   Edited.

For Appendix B:

1. "Kriging variogram modelling" is not a correct phrase.

We thank the reviewer for the suggestion. In the revised version "Kriging variogram modelling" is renamed to "variogram modelling".

2. P23L7 There is a comma too much before i

Corrected.

3. P23L9 The variogram function does not provide a unique solution for weights, there are steps in between.

We agree with the reviewer. In the revised version we have modified the sentence which reads as follows: "*The theoretical variogram function (γ \* (h)) allows the analytical estimation of variogram values for any distance and provides the unique solution for weights with intermediate steps required for kriging interpolation (for more details refer to Adhikary et al., 2015)".*

4. P23L23 remove "auto-"

Removed.

5. "precipitation" in the caption is at the wrong place

Corrected. Caption read as follows:

*Figure A2: Typical variogram model (a) and fitted variogram models for daily mean (b), 90th (c), 95th (d), and 99th (e) percentile precipitation, and number of wet days (f).*

Looking at Figs 2 and 3, I don't see that the degree in Eq 6 is actually divided by N-1.

We thank the reviewer for noticing the typo error in eq.6. We have corrected equation 6 in the revised version.

Check all references carefully. For example, I noticed that almost all citations on P12 are missing, the same for some citations on P18. Maybe there are also more? Villarini et al. (2008) is never mentioned in the text, maybe there are more extra references as well?

All the reference has been checked and corrected thoroughly.

Please check all **new** text carefully, that it grammatically correct and flows well. It is not as well written as the other parts of the manuscript. Some examples are found in the minor edits below.

Entire manuscript has been checked and corrected for all typos, grammatical error and flow.

A small detail on Fig 4 – are the stations in France really stations? This seems more like a gridded product to me, either interpolated from observations or a reanalysis product like ERA-INTERIM or ERA5.

We have added the following clarification: For parts of France precipitation data on a 0.22° x 0.22° rotated pole grid from E-OBS is used (Haylock 217 et al., 2008).

Haylock, M. R. et al. (Oct. 2008). A European daily high-resolution gridded data set of surface temperature and precipitation for 1950–2006". Journal of Geophysical Research 113. DOI: 10.1029/709 2008JD010201

P18L29-P19L8. What is referred to as intuitive is actually not necessarily correct. The kriging variance is independent of the local variability, as it is model based. It is dependent on the variogram (which is based on all the data) and the distance and direction to other observations. Hence, the sentence and the relationship to the figures is confusing. Then Heuvelink and Pebesma (2002) is not in the bibliography. I understand what is meant with the rest of the paragraph but it is not well written. It is wordy and incoherent. Please rephrase.

We again thank reviewer for highlighting the need to rephrase the paragraph. Considering the reviewer's suggestion edited section (P18/L22-28) read as follows:

*The results presented (Fig. 7) supports the conclusion derived from the kriging error analysis in section 4.4. Removing an influential station (Fig. 7b) fosters higher kriging errors than removing a random low ranking station (Fig. 7c). Hence, the new measure could support the optimal design of large hydrometric networks or redesign of existing hydrometric networks by ranking nodes. The influence of the similarity measure, number of stations present in the network, spatial boundary, data length, and threshold has to be further investigated before the method can become fully operational. Acknowledging the infancy state of complex network science in hydrology, we emphasize the need for more intensive application*

**Minor Edits**

P1L16 Remove comma after although. Then maybe add one after attention in next line, or revise word order.

Corrected.

*Although the design of hydrometric networks is a well-identified problem in hydrometeorology and has received considerable attention, yet there is scope for further advancement.*

P1L18 defined as A collection of nodes

Corrected.

*In this study, we use complex network analysis, defined as a collection of nodes interconnected by links, to propose a new measure that identifies critical nodes of station networks.*

P3L17 This addition to the sentence seems somewhat lost, as the first part of the sentence refers to importance of nodes, not use in hydrology (or meteorological observations). Should probably also be "in ITS infancy"?

Corrected.

*In this study, we propose a complex network-based method to identify the influential and expendable stations in a rainfall network. Several methods in the field of complex networks have been proposed to evaluate the importance of nodes (Chen et al., 2012; Hou et al., 2012; Jensen et al., 2016; Kitsak et al., 2010; Zhang et al., 2013 and Hu et al.,*

*2013), however, the application and interpretation of complex network in hydrology (or meteorological observations) is in its infancy state.*

P5L6-7 Threshold alpha percentile -> The alpha percentile threshold?

Corrected.

P5L20&P6L2 Maybe "stations" instead of "grid sites"

Corrected.

P7L7 Maybe add reference to figure: "in the networks N1 and N2 IN FIG 1, all nodes…"

Done.

P8L21-22 Kurths et al. (2019) – move comma! Next line it should probably be timescaleS

Corrected.

P9L4-8 The last part is an assumption. Additionally, the sentence is hard to read as it is. Spatially large variations -> large spatial variability? Across locations or between locations? And is "restricted" the right word here?

*As mentioned, global bridges connect different parts of a network (e.g. teleconnection between Indian rainfall and ENSO). Measuring and interpretation of large spatial variability, process identification, interpolation of measurements and transferability of precipitation measurements across locations, would be limited in the absence of high $B_i$ nodes.*

P10L7-8 "and among others" – what is meant here? Something missing?

Done.

P10L11-12 I think the sentence would be more correct by adding for: FOR process identification, FOR interpolation of measurements and FOR transferability … Across stations?

Done.

P13L10 comma before "which". Next line: whereas -> where?

Done.

P14L17-19. Sentence is not well written, please rephrase. (particularly: "for certain threshold range 90-99" and "robust for comparatively high threshold")

Corrected.

*To validate results, we performed analysis for 90-99th percentile threshold range and observe that node rankings are robust for high threshold.*

P16L9 "are removed terms as Knew" – something missing?

Corrected.

*Then, we measure the increase or decrease in the kriging standard error across the study area when stations are removed and that is termed as $K_{new}$.*

P16L10 remove comma after similarly

Done.

P16L14 we run THE model

Done.

P16L15 R is not defined.

Done.

P18L11-12. This sentence is incomprehensible. I don't understand what weighted kriging could do, "strengthen" should be "strengthened" and what does the last part mean?

We have removed this statement in the revised version.

[revised manuscript text omitted]

**B. Variogram modelling**

The kriging modelling assumes  a theoretical variogram function that is  fitted with an experimental variogram of the observed data. The experimental variogram ($\gamma(h)$) is calculated from the observed data as a function

5 of the distance of separation ($h$)  (Adhikary et al., 2015), and is given by

$$\gamma(h) = \frac{1}{2N(h)} \sum_{i=1}^{N(h)} \left[ (Y(i) - Y(j))^2 \right], \tag{A1}$$

where $N(h)$ is the number of sample data points separated by  distance $h$;  $i$ and $j$ represent sampling locations separated by  $h$; $Y(i)$ and $Y(j)$ indicate values of the observed variable $Y$, measured at the corresponding locations $i$ and $j$, respectively. The theoretical variogram function ($\gamma$-*-($h$)) allows the analytical estimation of variogram values for any distance and provides the unique solution for weights with intermediate steps required for

10 kriging interpolation ( Adhikary et al., 2015).

The variogram models are a function of three parameters:  the range, the sill, and the nugget (Fig. A2 (a)). The range is  the distance, where the models first flatten out, i.e., station locations within the range distance are spatially -correlated, whereas locations farther apart

 are not. The value of γ at the range is called the sill,  which is. estimated by the variance of the sample . The nugget represents measurement error and/or microscale variation at very small spatial scales   and is seen as a discontinuity at the origin of the variogram model. The ratio of the nugget to the sill is known as the nugget effect and may be interpreted as the percentage of variation in the data that is not related to spatial. The difference between the sill and the nugget is known as the partial sill (Adhikary et al., 2015; Keum et al., 2017).

The values of all parameters and the resulting variograms for daily mean, 90th , 95th , and 99th percentile precipitation and number of wet days are reported in Table A2 and Fig. A2 (b-d), respectively. The variogram has been kept constant during network reductions.

**(a) Typical variogram model**

[Figure]

**(b) Mean**

[Figure]

**(c) 90th percentile**

[Figure]

**(d) 95th percentile**

[Figure]

**(e) 99th percentile**            **(f) wet days**

[revised manuscript text omitted]